# GlyContact analyzes glycan 3D structures at scale

Luc Thomès [1], Roman Joeres [2,3,4], Zeynep Akdeniz[2,3] & Daniel Bojar [2,3] ✉

Glycans are branched, structurally diverse, and highly flexible biomolecules. These characteristics make glycoanalytics and structural characterization challenging, resulting in often unclear structure-to-function relationships. GlycoShape, currently the largest open-access database of glycan 3D structures from molecular dynamics (MD) simulations, provides an opportunity to fill this information gap. Here, we present GlyContact, an open-source Python package designed and developed to retrieve, process, and analyze glycan 3D structures, from MD, NMR, or X-ray crystallography. We demonstrate that GlyContact can (i) unveil the impact of sequence context on glycan motif structure, (ii) yield a predictive understanding of motif flexibility and surface accessibility on lectin-glycan binding, which improved lectin-binding prediction by ~ 7%, and (iii) accurately predict torsion angle distribution between disaccharides using von Mises graph neural networks. We envision that GlyContact will allow researchers to explore glycan structures within their 3D space, obtaining insights into their biological functions. GlyContact is available open-access at https://github.com/lthomes/glycontact.

Glycans are complex carbohydrates that play essential roles across all domains of life[1]. These diverse structures are involved in countless biological processes, from cell signaling and immune response to protein folding and cellular recognition[1,2]. Found on cell surfaces, attached to proteins, and integrated into cellular matrices, glycans are key components of the molecular machinery that enables life.

Biological functions emerge from molecular interactions, which are, in turn, mediated by a combination of three-dimensional (3D) compatibility, dynamic behavior, and appropriate physicochemical properties. Within this framework, the enzymatic modification of glycans, their recognition by glycan-binding proteins (or lectins), and interactions with other biomolecules all require specific 3D arrangements existing within precise conformational equilibria[3].

Glycan structures are known to modulate biological function, where changes in sequence and branching have been demonstrated to facilitate or hinder specific events. This ability is enabled by changes in the 3D structure and dynamics of glycans, which are directly and uniquely determined by their sequence and branching. The subtleties of these conformational equilibria are not easily accessible through experiments but can be predicted and analyzed by molecular dynamics (MD) simulations, within the constraints of sufficient deterministic, or stochastic sampling[3]. For example, previous work has shown that the addition of β2-Xyl to the central Man in the core pentasaccharide of plant N-glycans shifts the conformational equilibrium of the α6 arm, relative to the non-xylosylated form, resulting in an extended arm in the dominant conformation at equilibrium[4]. Understanding these structure-function relationships can be crucial to better understand glycobiology and to design glycomimetic compounds for therapeutic or diagnostic purposes.

Glycan 3D structures destined to MD simulations can be built from scratch with carbohydrate builders such as GLYCAM-Web[5,6], CHARMM-GUI[7], and MatrixDB, the latter developed specifically for GAGs[8], while MD-based applications are also available for evaluating the effects of glycan shielding on protein surfaces, e.g., with GlycoSHIELD[9]. The recently developed GlycoShape[10] (https://glycoshape.org) is a unique database and toolbox, providing 3D

[1]University Lille, CHU Lille, ULR 7364 - RADEME - Maladies RAres du DÉveloppement embryonnaire et du Métabolisme, Lille, France. [2]Department of Chemistry and Molecular Biology, University of Gothenburg, Gothenburg, Sweden. [3]Wallenberg Centre for Molecular and Translational Medicine, University of Gothenburg, Gothenburg, Sweden. [4]Saarbruecken Informatics Campus, Saarland University, Saarbruecken, Germany. ✉e-mail: daniel.bojar@gu.se

structural information on glycans at equilibrium and bespoke software applications designed to functionalize proteins with the appropriate glycan structures, offering the glycobiology community unprecedented access to structural information and thus opening possibilities for studying glycan structure-to-function relationships.

To assist and complement the rise in the availability of glycan 3D structural data from MD-based libraries, such as GlycoShape, and from experiment, we believe that the scientific community would benefit from a software specifically designed to process glycan 3D structures, extract key conformational parameters, and link those to functional biological traits in combination with machine learning approaches. While tools such as Privateer[11,12] exist for some of these purposes, these are geared towards structural biology applications, particularly to validate and/or correct glycan structures obtained from X-ray crystallography, and are harder to natively connect to the sequence-based glycoinformatics world[13].

Here we introduce GlyContact, an open-source Python package designed specifically for glycan structure analysis that can be entirely operated with glycan sequences (in any chosen nomenclature). GlyContact enables researchers to easily retrieve and process 3D glycan structures, perform sophisticated conformational analyses, and investigate structure-function relationships. Our package for instance includes approaches for analyzing relationships between glycan composition/sequence/class and structural properties, quantifying 3D flexibility, evaluating surface accessibility for protein interactions, and using (as well as predicting) these features in glycan-focused AI approaches.

To illustrate the usefulness of GlyContact, we demonstrate its capabilities and applications through multiple case studies. We show that (i) sequence context influences structural properties of motifs in a systematic manner, (ii) motif surface accessibility explains variance in glycan-binding proteins such as lectins, (iii) adding glycan 3D information improves state-of-the-art glycan-AI prediction models independent of improved protein representations, and (iv) conformer-specific glycan torsion angles can be predicted from sequence alone via an AI-driven von Mises mixed model, which we establish here, that was trained on almost 7000 3D conformers. These examples illustrate how GlyContact can be used to extract insights from already existing structural data into glycans and their biological functions, such as binding to lectins.

## Results

### GlyContact aggregates and calculates structural properties of glycans across conformers

With the increasing availability of glycan 3D information, e.g., via NMR and MD, it finally becomes conceivable to probe structure-function associations for complex carbohydrates, such as analyzing the general effect of motif surface availability on lectin binding. Yet, currently, this type of analysis—bridging the realm of macromolecular coordinates and intricate, sequence-based motifs—is not yet readily available to researchers. Further, most glycans are structurally disordered, existing at equilibrium within structural ensembles, rather than as a single 3D structure, that can count many (largely) different conformers. To overcome these hurdles and link the rich information in glycan 3D structures to the deep analytic capabilities offered by glycoinformatics software such as glycowork[14], we thus set out to develop a series of data processing and analysis pipelines specifically geared towards glycan 3D structures, resulting in the open-source Python package GlyContact (Fig. 1a).

At its core, GlyContact enables the systematic investigation of glycan conformational properties by linking glycan 3D information and glycan sequences. While users can always provide file paths to specific PDB files, the default mode of GlyContact is for users to input glycan sequences, with internal functions mapping those to the correct conformer structures from, e.g., GlycoShape[10]. Most of the

functions within GlyContact rely on the *annotation_pipeline* workflow, which gathers relevant PDB files for conformers and then extracts and formats coordinates and links between monosaccharides. GlyContact, by default, will automatically select physiologically relevant conformers, such as reducing-end α-conformers for *O*-glycans and β-conformers for *N*-glycans. Next to MD data, GlyContact further supports processing glycans from lectin-glycan complexes (e.g., UniLectin3D[15]), as well as covalently linked glycans from glycoproteins within regular PDB files (e.g., from NMR or X-ray crystallography). To maximize accessibility and user-friendliness regarding file-parsing, no changes in workflows are necessary to specify the type of file (MD, NMR, X-ray, glycan, protein-glycan complex, covalent glycosylation, etc.), as the file structure and necessary actions will all be automatically inferred and executed.

This can then be followed by a wide range of analyses, as presented below, from directly analyzing coordinates between conformers over calculating biophysical properties, such as surface accessibility, to structural alignments between glycans. Most of the functionalities within GlyContact can be retrieved and used in three ways: (i) direct tabular results for export, (ii) plotting via heatmaps or overlays of glycan sequences, and (iii) mapping properties such as conformational parameters onto glycan graphs resulting from glycowork, e.g., to be used in downstream AI or data science applications.

While users can of course supply their own glycan PDB files for all functions, GlyContact was initially developed to analyze glycan conformers retrieved from the rapidly growing GlycoShape database[10], for which GlyContact can successfully extract and process 98.7% of the structures of its 717 glycans with deposited PDBs (Supplementary Data 1), which correspond to 6911 conformer 3D structures. Using the cluster proportions of these glycan conformers, obtained from MD simulation, then, e.g., allowed us to study glycan geometry via distance-based analyses. By calculating all pairwise monosaccharide (or atom) distances within each conformer and aggregating this information in a weighted manner across conformers, GlyContact can generate both contact maps and changes between contacts that reveal local and long-range interactions during the flexible rearrangements in solution (Fig. 1b, Supplementary Fig. 1).

In general, GlyContact offers excellent performance, with most glycan structures (i.e., all their conformers) being processed in less than 0.1 s on a regular laptop (Supplementary Fig. 2). To ensure that other types of glycan structures can be correctly processed as well, we also report that 1437 lectin-glycan complexes from UniLectin3D[15] can be correctly processed (Supplementary Data 2), with the remaining cases exclusively comprising heavily chemically modified, synthetic glycans. We then stored these extracted glycan structures from UniLectin3D in GlyContact (*glycontact.process.unilectin_data*), and ensured that functions could automatically retrieve structures for glycans that were not yet present in GlycoShape from there.

To understand how glycan structure influences potential interactions with proteins, we next implemented the calculation and visualization of surface properties of carbohydrate 3D structures. GlyContact can compute the solvent-accessible surface area (SASA) for all monosaccharide residues separately and maps these SASA values onto IUPAC-condensed glycan sequences, as well as glycan graphs, which can then—thanks to the seamless integration with glycowork—be visualized via GlycoDraw[16] (Fig. 1c). We further note that GlyContact benefits from the Universal Input platform that glycowork has introduced[17], allowing users to write glycans in any nomenclature of their choosing (e.g., IUPAC-condensed, IUPAC-extended, WURCS, GlycoCT, GLYCAM, Oxford, GlycoWorkbench, LinearCode®, CSDB-linear, etc; as well as common names, such as 'LacNAc' or 'sucrose'), without the need for format conversion.

Calculating SASA, together with similar operations for characteristics such as flexibility (Fig. 1d), revealed stark heterogeneity of even the same monosaccharide in different positions of the glycan, which is

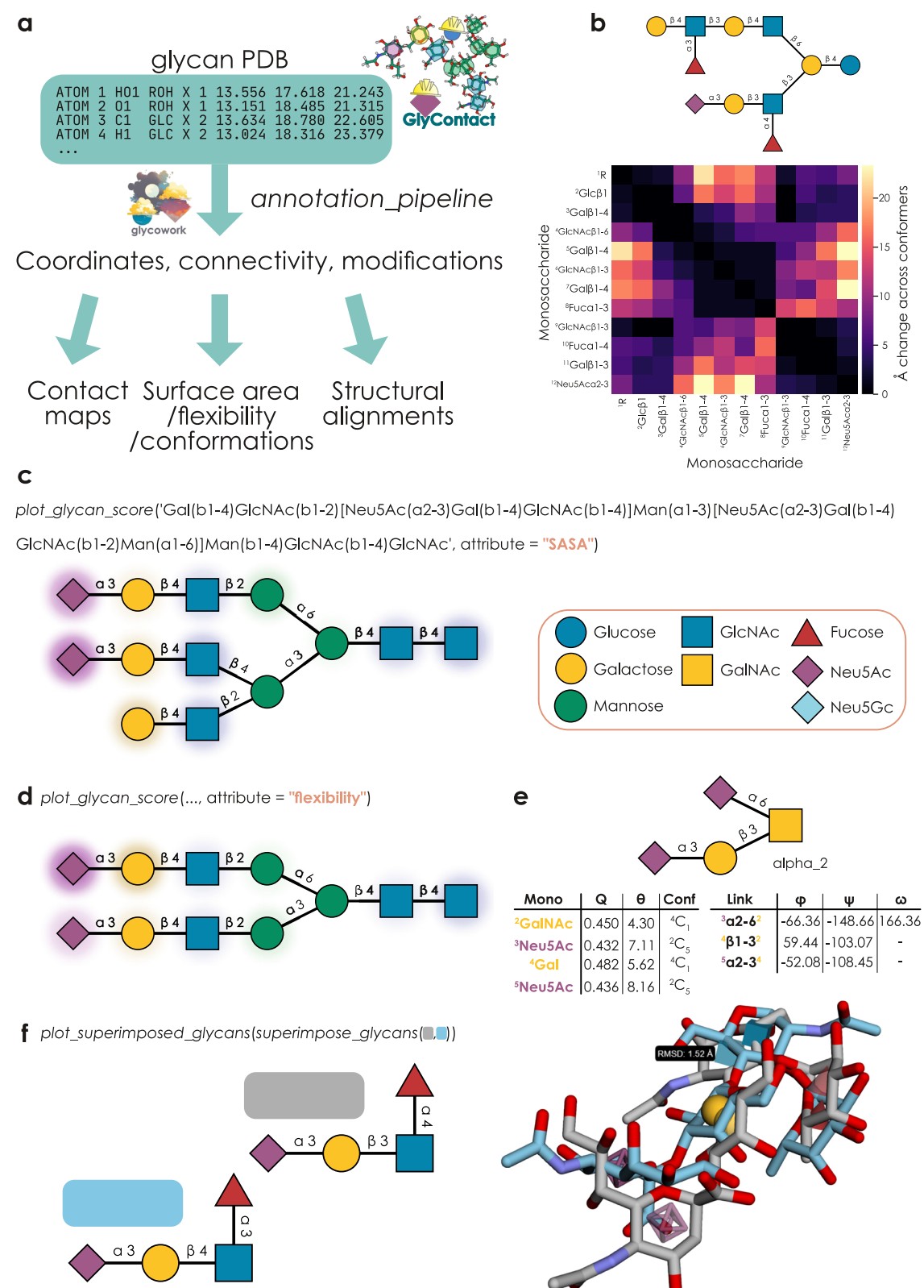

hidden in the 'flat' textual representation of glycans. For MD libraries (GlycoShape), we estimated monosaccharide-level distance-based flexibility as the movement between conformers, while we used B-factors in the case of X-ray crystallography, converting both cases to root-mean-square fluctuation (RMSF) for comparability. We note here that B-factor derived RMSF values should only be used to compare glycans within the same crystal structure, not between different

structures, as B-factor distributions can vary between PDB files. Further, we note that MD conformers are aligned along their first five residues and thus distance-based flexibility will likely underestimate the flexibility of residues close to the reducing end. Therefore, lastly, we also calculated torsion-based flexibility (the weighted average of torsion angle spread) as an approach to assess flexibility that does not suffer from the limitation of having to first align parts of the structure.

**Fig. 1 | GlyContact is a powerful platform to process and analyze glycan 3D structures across multiple conformers. a** Schematic overview of the main capabilities of GlyContact. Aided by functionality from glycowork[15], GlyContact extracts glycan-relevant information from PDB files and then can be used to analyze coordinates within and across structures, as well as calculate biophysical parameters, from solvent-accessible surface area (SASA) to monosaccharide conformations, as well as glycan 3D alignments. **b** Monosaccharide contact map of example glycolipid. Across all beta conformers of the shown glycan, using the *draw_contact_map* and *inter_structure_variability_table* functions, we compared how stable the contacts between two monosaccharides were across conformers. **c** Calculating monosaccharide-level SASA values. Using the *plot_glycan_score* function, we calculated SASA values for all monosaccharides of the shown glycan (weighted and averaged across its beta conformers), mapped the values to the glycan graph, and visualized them via GlycoDraw[17] as normalized SASA values (0-1) in the form of a color gradient. **d** Calculating monosaccharide-level flexibilities. Similar to (c), we used the *plot_glycan_score* function to visualize the distance-based flexibility, weighted across beta conformers, of each monosaccharide for an example structure. **e** Conformational and torsion angle analysis of example structure. For a single alpha-conformer of the disialyl-T antigen, we used the *get_ring_conformations* function to obtain ring puckering (Q), polar angle (θ), and ring conformations (Conf) for each monosaccharide and the *get_glycosidic_torsions* function to obtain the torsion angles (φ, ψ, ω) for all disaccharide connections. **f** Aligning glycan structures. Using the *superimpose_glycans* and *plot_superimposed_glycans* functions, we visualized the best alignment between all conformers of sialyl-Lewis A (blue) and sialyl-Lewis X (gray), including their root mean square distance (RMSD) in Å. All glycans in this work have been drawn with GlycoDraw[17] in the Symbol Nomenclature For Glycans (SNFG).

Importantly, all these calculations are done 'on-the-fly' via highly optimized workflows in less than a second, all the way from raw.pdb files, and combined across different conformers in the case of MD (weighted by cluster proportion), allowing for ready re-analysis of structures.

Overall, this approach confirmed that terminal residues exhibit both higher accessibility and higher flexibility, on average, than core residues, which we exemplified with the monosaccharide galactose (Supplementary Fig. 3). These differences in accessibility patterns may help explain the distinct binding preferences of different glycan-binding proteins, or lectins, as shown below.

Next, we were interested in analyzing the internal geometry of glycan structures more formally and, analogous to our SASA/flexibility operations, calculated and mapped monosaccharide properties (ring puckering, polar angle, chair conformation) as well as linkage properties (torsion angles; Fig. 1e), which again can be used to analyze the structural diversity of glycans as well as featurize them for downstream applications, such as informing glycan-focused AI models[18–20]. One example for this can be found in recent approaches to capture glycan dynamics via non-linear torsion-torsion correlations[21], which we also support here via the *analyze_torsion_torsion_correlations* function to yield insights into structural constraints and dynamics (Supplementary Fig. 4). For the case of glycoprotein inputs, this functionality also returns the torsion angles between the reducing end monosaccharide and the amino acid at the glycosylation site, as well as the flexibility of this amino acid.

Applying these functionalities to the whole GlycoShape database then allowed us to compare monosaccharides across thousands of occurrences, which generally showed similar ranges of puckering for commonly investigated monosaccharides (Supplementary Fig. 5), and to rapidly calculate and visualize Ramachandran plots for common disaccharide torsion angles (Supplementary Fig. 6), matching torsion angle ranges reported by others[12,22].

Lastly, inspired by the popular application of protein structure alignments, we added the functionality of geometric point cloud alignment to GlyContact. Using the *superimpose_glycans* function with two glycan sequences will automatically find the two conformers that form the best-matching alignment between these glycans, which can then be visualized via the *plot_superimposed_glycans* function that results in a rotatable/interactive 3D alignment with both the chemical depiction of glycans as well as the visual guidance of the 3D-SNFG icons[23] (Fig. 1f). This can then also be parallelized to find the best matching molecule (from all existing glycan structures) to a single glycan of interest via the highly optimized *get_similar_glycans* function.

## Revealing systematic structural impacts of glycan motifs

Our performant and modular approach to glycan superimposition or structural alignment then paved the way for a systematic comparison of glycan structure similarity. For this, we pairwise aligned all GlycoShape glycan structures (via an SVD-based Kabsch algorithm using

k-d spatial trees; see Methods) and then clustered glycans via the resulting alignment distance matrix (Fig. 2a, Supplementary Fig. 7). This resulted in a recognizable clustering by glycan class, yet we would like to point out that this was mostly a function of glycan size/branching (e.g., *N*-glycans being, on average, larger than *O*-glycans), which significantly correlated with the spread on the x-axis (Pearson's $r = 0.60$, $p < 0.001$) and explained 47% of the total variance (Supplementary Fig. 7). In addition, we observed a distinct spread of glycan structures on the y-axis as well, yet none of the sequence or structural metrics (e.g., average SASA) that we tested correlated substantially with this spread (i.e., Pearson's $r < 0.15$). This could indicate yet unconsidered levels of structural similarity between glycans that are not apparent on the sequence level.

We next set out to build workflows to analyze the systematic impact of sequence modifications on glycan structure, which would allow researchers a certain intuition about the expected structural properties of a glycan. Probing this generally is very difficult, due to the plethora of confounding factors (e.g., naively comparing glycans with and without core fucose could result in different sequence lengths and contents being compared). A careful comparison is also needed because the addition of the motif itself trivially changes the glycan properties at the site of addition, whereas we aimed to identify distal impacts of motif addition on glycan structure.

Therefore, we established the *find_difference* workflow, in which we collected 'twins' (i.e., glycans differing only in the characteristic to be studied), and excluded the motif-of-interest from the actual comparative analysis. Applying this to core-fucosylated *N*-glycans (Fig. 2b), we report that the addition of core fucose, on average, lowered the available surface area of a glycan, indicating a tighter packing. We note that this method analyzes residuals between twins and thus controls for all potential confounders in other sequence characteristics. In the case of core Fuc, we for instance report that subgroup analyses of only α2,3 or only α2,6 linked Neu5Ac-containing glycans resulted in the same conclusion (Cohen's $d_z = -1.99$ and $-1.56$, respectively). We caution that, while this workflow can tease out general and systematic impacts of motif effects on glycan structure, there will likely always be additional sequence-specific effects of a motif as well. We then applied this same analysis approach to plant *N*-glycans, supporting earlier observations[4] that the introduction of β2-Xyl in the trimannosyl-core 'opens up' the overall structure, leading to a higher average SASA (Supplementary Fig. 8a), and revealed that Neu5Ac-containing glycans, on average, exhibited higher distal flexibility than their Neu5Gc-containing 'twins' (Supplementary Fig. 8b).

Next, we wanted to reverse this set-up and ask the question of whether distal sequence context would alter the properties of glycan motifs themselves (i.e., is a sialyl-LacNAc motif always a sialyl-LacNAc motif, regardless of context). For this, we analyzed whether the same sequence motif in different glycan classes (*N*-linked/*O*-linked/glycolipid) would exhibit different structural properties. Indeed, many motifs were characterized by such an impact of sequence backbone, for

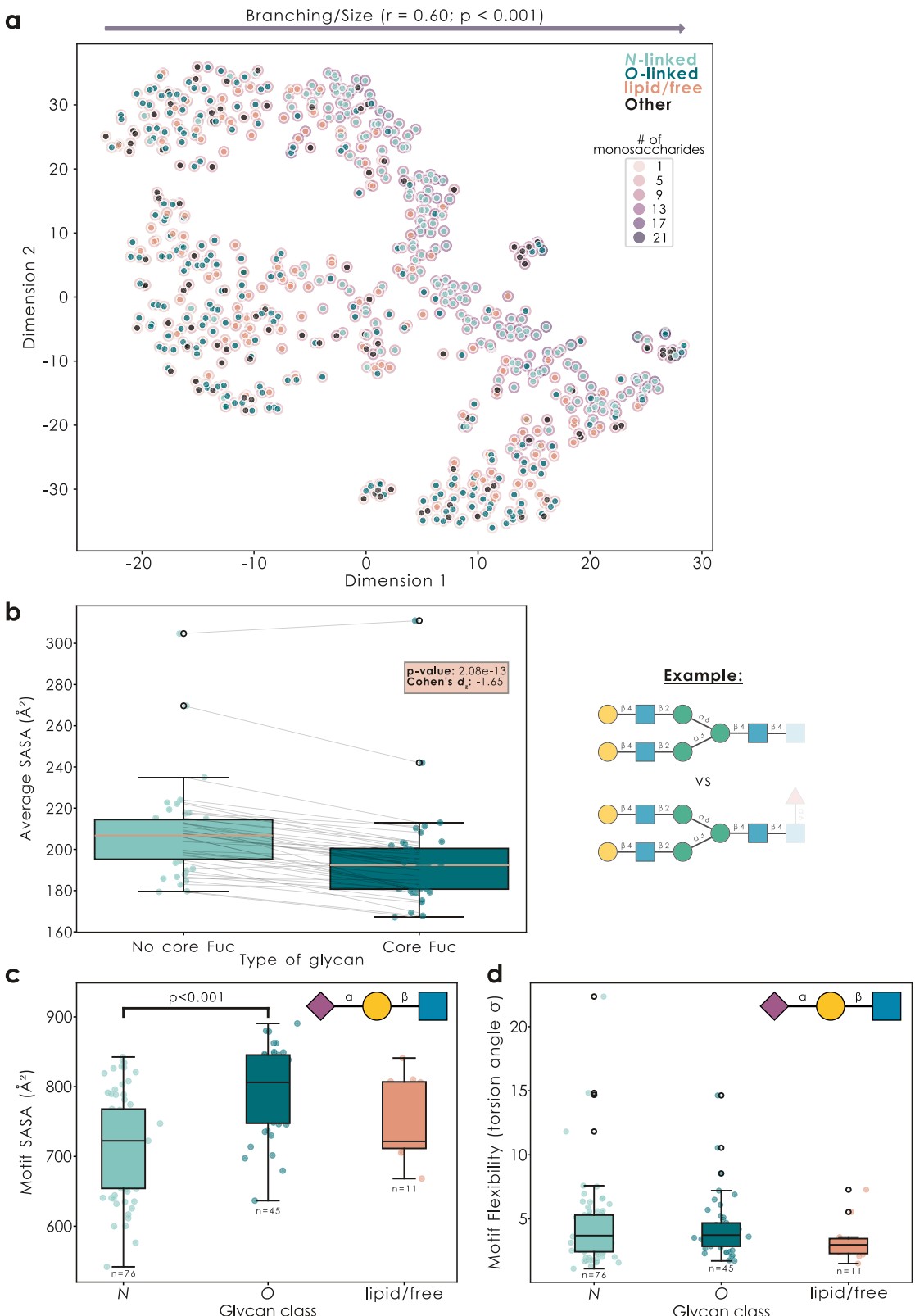

instance with sialyl-LacNAc motifs being less surface-exposed in *N*-glycans than in any other glycan type (Fig. 2c and d). We confirm that the exact same effect was also observed with Neu5Gc-containing sialyl-LacNAc motifs, further supporting this observation (Supplementary Fig. 9a and b). However, we stress that this type of analysis was conducted on isolated glycans from GlycoShape data and it is to be expected that the protein backbone will impact these properties. Even

between *N*- and *O*-glycans, this isolated analysis should not be seen as reflecting the glycoprotein situation (e.g., majorly influenced by the flexibility of disordered protein regions in the case of *O*-glycans), but rather only the marginal contribution of the glycan sequence context.

To illustrate that the exact effect of distal sequence contexts was motif-dependent, we then went on to show that the Sd[a] motif, Neu5Acα2-3(GalNAcβ1-4)Gal, exhibited higher surface-exposure in

**Fig. 2 | Finding structural patterns at scale with GlyContact. a** Glycan 3D structures form distinct clusters. For all glycans on GlycoShape ($n = 717$), we used the *superimpose_glycans* function from GlyContact, with the flag "fast=True", to calculate all pairwise alignments (always with the best-matching pair of conformers) via an SVD-based Kabsch algorithm using $k$-d spatial trees. Alignment distances were then used to cluster glycans via t-SNE initialized with a PCA. Glycan class is indicated by color and glycan size (i.e., number of monosaccharides) by a colored halo. The correlation of glycan size with x-axis spread is provided as Pearson's r, with the results of a two-tailed t-test for testing the correlation against zero ($p = 0.0004$). **b** Glycan epitopes affect distal structural properties. For the example of core fucose in *N*-glycans, we used the *glycontact.visualize.find_difference* function to gather all 'twins' (pairs of sequences that only differed in the presence/absence of core fucose) for which we had structural data from GlycoShape ($n = 43$) and compared their average SASA values (excluding core fucose and its attachment site). Results are shown as box plots (line indicating the median, box edges

indicating the 25th and 75th percentile, whiskers indicating the 95% confidence interval, and black circles indicating outliers), as well as an overlaid scatter plot of the actual values, with added horizontal jitter for visibility. Statistical testing involved a paired two-tailed t-test and Cohen's $d_z$ as an effect size for paired samples. **c, d** Glycan motifs exposed by different glycan classes exhibit different structural characteristics. For the various isoforms of sialyl-LacNAc (Neu5Acα2-? Galβ1-?GlcNAc) in GlycoShape glycans, we used the *glycontact.visualize.find_difference* function to sum the monosaccharide-level SASA values for each motif (**c**, $p = 0.00001$) and averaged their flexibility (**d**). Then, we grouped glycans by glycan class (using the *glycowork.motif.processing.get_class* function) and analyzed differences in SASA (**c**) or torsion-based flexibility (**d**) across classes by a one-way ANOVA, followed by Tukey's Honestly Significant Difference post-hoc test. The number of analyzed motif instances is provided under each box plot. Data are depicted as mean values, with box edges indicating quartiles, and whiskers indicating the remaining data distribution up to the 95% confidence interval.

*O*-linked glycans, as well as no difference in torsion-based flexibility (Supplementary Fig. 9c and d). Overall, we argue that this potential effect of distal sequence context needs to be considered separately for each motif, yet could be powerful in explaining well-known lectin preferences for a given motif in a given glycan class[24], as well as combine with the phenomenon of presentation of a binding motif, which is known to affect lectin binding[25].

## Motif flexibility and surface accessibility explain lectin binding

Following up on this concept that motif presentation is influenced by distal sequence context and that this presentation, in turn, might enable or preclude a lectin to bind to a motif, we next set out to probe whether structural properties of binding motifs could explain the binding behavior and preferences of lectins. For this, we started by analyzing a set of commonly used lectins with well-characterized binding preferences[24] (Supplementary Data 3). With the example of the popular lectin RCA-I (binding terminal LacNAc motifs), we first found that, among glycan microarray data, not all glycans with terminal LacNAc motifs were bound by RCA-I, which we then attempted to explain with the structural properties of these LacNAc motifs (Fig. 3a and b). Indeed, amongst glycans with terminal LacNAc chains, the surface availability of the RCA-I binding motif positively correlated with its likelihood of being bound, explaining an additional 15% of the observed variance in binding.

We would like to emphasize that this dependency is a lectin-specific property and it cannot be assumed a priori that all lectins generally prefer highly available motifs. For the example of the widely used Neu5Acα2-6 binding lectin SNA (Fig. 3c and d), we for instance report no association of binding with the surface availability of the sialic acid, yet a strong preference for high distance-based flexibility (Supplementary Fig. 12b), possibly because of the known preference of SNA for its binding motif on *N*-linked glycans[26]. On the other hand, lectins such as the pathogen-derived, fucose-binding BambL from *Burkholderia ambifaria*[27], exhibited a strong preference for surface-available versions of their binding motif, as well as a significant dependence on torsion-based motif flexibility (Fig. 3e and f).

Our overall conclusion here is that most lectins have some further structural specification that dictates their binding, next to a sequence stretch, leading us to the statement that lectins do not bind sequences but, rather, specific 3D contexts. Other examples included the popular lectin AAL, which not only required fucose for binding, but actually surface-available Fuc residues (Supplementary Fig. 10a and b), resulting in the insight that many glycans contain terminal fucose moieties, yet fewer glycans actually carry an AAL binding motif. Similarly, not all terminal core 1 epitopes (Galβ1-3GalNAc)—but only surface-available ones—were actively bound by PNA (Supplementary Fig. 10c and d). Lastly, we emphasize that not all lectins followed the simple rule of preferring higher surface-availability, as lectins such as PHA-L actually

showed a trend of preferring their binding motif to be somewhat buried (Supplementary Fig. 10e and f), potentially indicating additional interactions with distal sequence contexts. In addition to SASA and flexibility, we also analyzed hydroxyl group conformation and observed that several lectins had a significant dependency on the equatorial/axial positioning of hydroxyl groups in their binding motif(s) (Supplementary Fig. 11), offering another angle to explain lectin-glycan binding in a data-driven manner.

We next aimed to systematically probe the binding behavior of lectins and grouped 58 well-known lectins according to their dependency on motif surface availability and flexibility (Fig. 4a, Supplementary Fig. 12, Supplementary Data 3). This revealed some general tendencies in lectin behavior, such as that most lectins had some requirement for surface-availability of their binding motif (though highly variable as to the strength of this constraint), whereas the dependency on motif flexibility was more split, with approximately half of the lectins preferring rigid motif contexts (e.g., HPA) and the other half (e.g., UEA-I) flexible binding motifs, with a weak tendency for preferring more rigidly presented motifs if SASA was important for binding (Pearson's $r = -0.08$ with torsion-based flexibility, $-0.42$ with distance-based flexibility). We caution that the effects we find here, while potent in our efforts to explain glycan array data via structural characteristics, might differ in physiological settings, since protein-linked glycans will be influenced in their structural characteristics (e.g., flexibility) by the protein backbone they are attached to. Here, we envision that lectins binding to non-reducing end terminal motifs, such as sialic acid-binding lectins, will match more closely with our findings than lectins binding to residues close to the reducing end (i.e., closer to the protein backbone). For example lectins exhibiting positive (AAL) and negative (CD33) correlations with the SASA values of their binding motifs, we have also validated this preference using experimental co-crystal structures of lectin-glycan complexes from UniLectin3D (Supplementary Fig. 13).

Reasoning that these structural preferences of lectins meant that a lectin typically bound a motif in only one (or few) structural configuration(s), we turned to the data we extracted from UniLectin3D, to study the effect of conformational selection in lectin-glycan interactions[28]. Here, we note that data from GlycoShape and Uni-Lectin3D serve different purposes. Whereas GlycoShape ensembles present an overview of energetically accessible conformations in solution, bound conformers in UniLectin3D yield insight into lectin-glycan interactions and present a specifically selected conformation from the overall ensemble. We also note that GlyContact has dedicated analysis routines for these cases, such as the *glycontact.process.get_glycan_sequences_from_pdb* function, to extract all IUPAC sequences of glycans in a PDB structure, or *glycontact.process.get_binding_pocket* to extract all protein residues/atoms that are in close vicinity to the bound glycan (Supplementary Fig. 14).

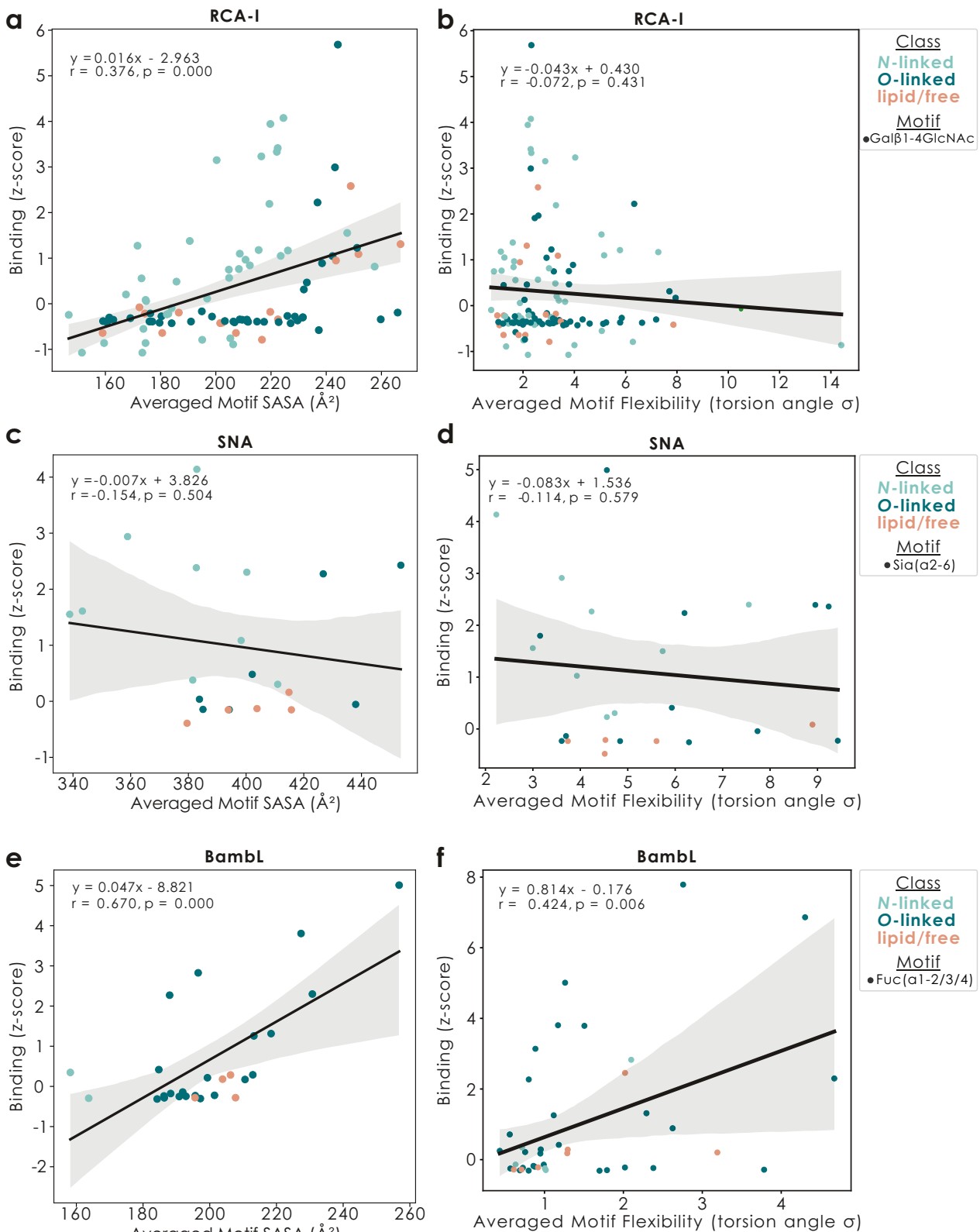

**Fig. 3 | Structural motif properties affect lectin binding.** For the lectins RCA-I (**a**, **b**), SNA (**c**, **d**), and BambL (**e**, **f**), we used the z-score transformed binding data from glycowork (v1.6) and correlated it with either the averaged SASA (**a**, **c**, **e**) or torsion-based flexibility (**b**, **d**, **f**) of the literature-known binding motif in each glycan that (i) carried the binding motif, (ii) had binding data, and (iii) was deposited on GlycoShape. On top of the data points as a scatter plot (colored by glycan class), we then drew a linear least-squares regression line (inner line depicting the mean and including 95% confidence band) as well as the regression equation, the Pearson's correlation coefficient r, and the p-value of a two-sided t-test of the regression coefficient against zero.

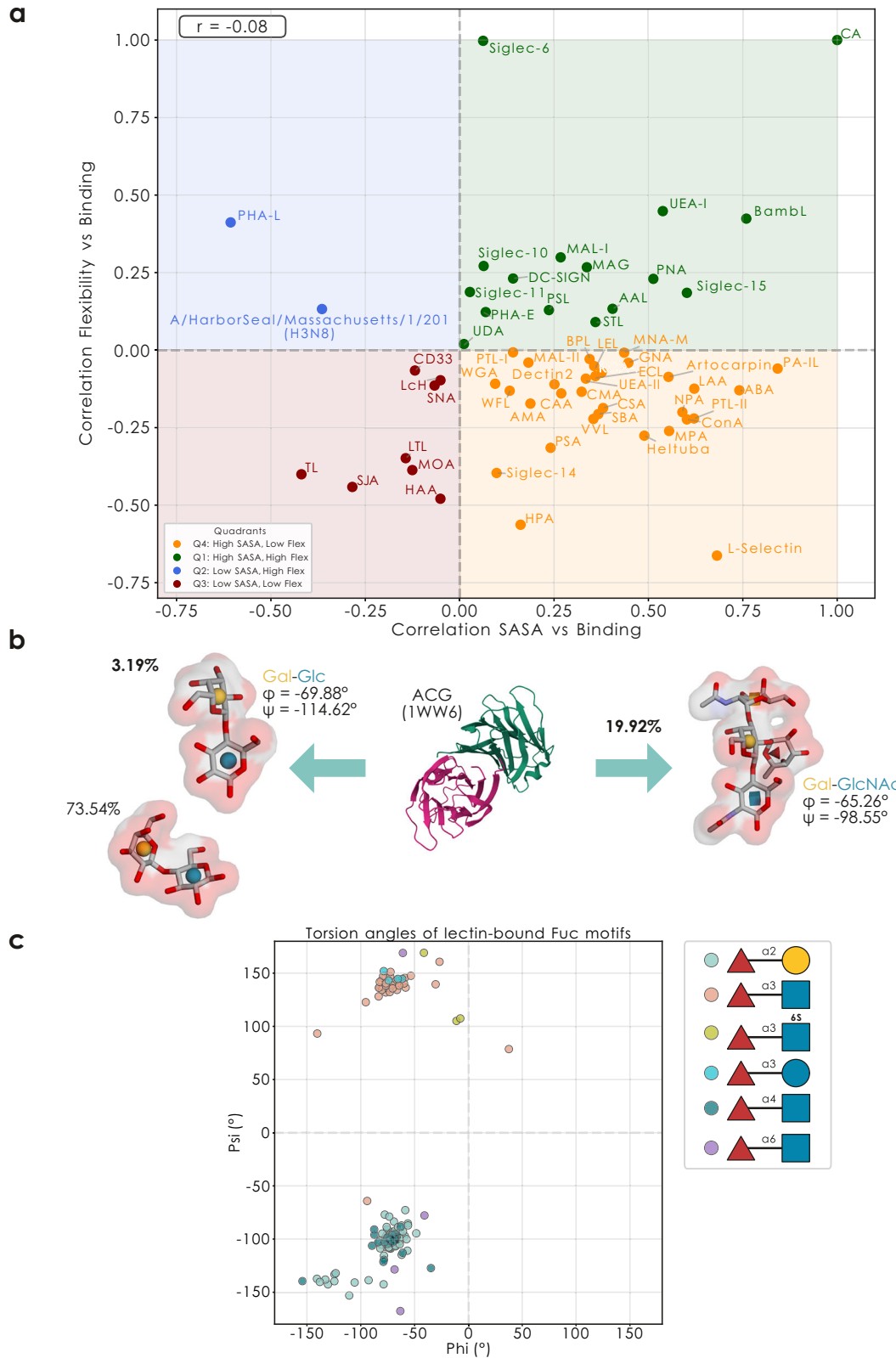

**Fig. 4 | Lectins can be grouped into structural binding preferences. a** Systematic analysis of lectins by their mode of binding. Each lectin is shown via their correlation of binding with average motif SASA and torsion-based flexibility, with quadrants (colored) loosely grouping the lectins. A Pearson correlation coefficient captures the relationship between SASA-dependency and torsion-based flexibility-dependency across lectins. **b** Conformer selection with the example of the fungal galectin ACG (depicted PDB: 1WW6). The bound lactose and blood group A conformers (as well as the most common lactose conformer) are shown with their van-

der-Waals surface, along with their conformer frequency. For bound conformers, the torsion angles for the Galβ1-4Glc (3WG1 [https://www.rcsb.org/structure/3WG1]) and Galβ1-4GlcNAc (3WG3 [https://www.rcsb.org/structure/3WG3]) linkage are shown too. **c** Ramachandran-like plot of the torsion angles of lectin-bound Fuc motifs from UniLectin3D. Using the *glycontact.process.get_glycosidic_torsions* function, we extracted φ and ψ angles from 133 co-crystal structures of lectins binding a glycan that contained fucose. Dots are colored by disaccharide sequence.

Using the fungal *Agrocybe cylindracea* proto-galectin (ACG)[29] as an example, we can see that lactose-bound structures here exhibit the rarest conformer of lactose (3.19%; e.g., 3WG1, Supplementary Data 4), whereas blood group A antigen-bound structures (the preferred ligand) interacted with the most common conformer of this epitope (19.92%, Fig. 4b; e.g., 3WG3, Supplementary Data 4). This, in conjunction with the relative lack of actual lactose-binding in practice, as measured via glycan array binding z-scores (1.17 for blood group A vs 0.41 for lactose, for the wildtype protein), indicates that ACG only exhibits affinity for a rare conformer of lactose. To explain this, we analyzed the conformation of the galactose environment in this rare lactose conformer and realized that it matched that of the galactose in the most common blood group A conformer (Fig. 4b), potentially explaining this binding event via conformational selection of a specific structural context.

Extending our analysis to more co-crystal structures, we next analyzed the observed torsion angles of fucose-containing binding motifs across all UniLectin3D structures (Fig. 4c, Supplementary Data 5). This revealed that, in its interactions with lectins, fucose is presented in two conformational clusters, represented by Fucα1-3GlcNAc, Fucα1-3GlcNAc6S, Fucα1-3Glc on the one hand (-φ/+ψ), and Fucα1-2Gal, Fucα1-4GlcNAc, Fucα1-6GlcNAc on the other (-φ/-ψ).

## Machine learning can use and predict glycan 3D structure properties

Given previous research and our findings here that glycan 3D presentation affects the efficacy of lectins binding their cognate binding motifs, we reasoned that inclusion of such information would improve current state-of-the-art deep learning models for predicting lectin-glycan interactions[30,31]. Since we did not have access to glycan 3D structures for all glycans that we have gathered lectin binding data for, we had to formulate a dedicated data-subset of the 368 glycans for which we had both array-based binding data and 3D structures. This resulted in a dataset of 354,958 unique binding z-scores (368 glycans, 2097 proteins; Supplementary Data 6). Then, as a baseline, we trained a typical LectinOracle model (ESM-C 300 M + SweetNet[30,32]) on this dataset to predict binding z-scores, given glycan and protein sequences, which resulted in competitive performance (MSE: 0.598).

Next, we added pre-computed 3D glycan features (SASA, flexibility, ring puckering Q, polar angle for monosaccharides θ, torsion angles for linkages) as additional node features into our glycan graphs and added those features as part of the graph convolutional neural network, resulting in LectinOracle$_{struct}$ (Fig. 5a), with a 7.4% superior performance (MSE: 0.557) to the base LectinOracle model, despite having very similar parameter counts. We note that this difference in performance cannot be explained by changes in parameter counts, as simply increasing the node features of the standard LectinOracle did not yield a performance boost (MSE: 0.596). For context, this gain in performance was comparable in order-of-magnitude to choosing a more performant protein encoder in our recent work on improving glycan-AI models[31]. Further, while a larger protein encoder, such as ESM2, did increase overall prediction performance, it did not remove or decrease the relative gap between LectinOracle and LectinOracle$_{struct}$, leading us to suggest that glycan structural information cannot be compensated via an improved protein representation.

To illustrate the effectiveness of this, we applied this LectinOracle$_{struct}$ model to the 58 well-known lectins characterized above, for which we have shown impacts of the surface-availability of their binding motifs (Figs. 3–4, Supplementary Fig. 10). As expected, making this information available to AI models such as LectinOracle improved the match between predicted and measured binding values (Fig. 5b), indicating the promise of including glycan 3D information into future models. We also note that the only lectins not benefitting from this inclusion tended to require high motif flexibility (Fig. 4a),

indicating their reliance on rare conformations for productive binding. We note that it remains to be seen whether these performance gains translate to predicting binding to protein-linked glycans, given the influence of protein structure on glycan structural properties that is not being captured in molecular dynamics-derived glycan structures in isolation. Lastly, we caution that using LectinOracle$_{struct}$ requires having access to glycan 3D information, which currently limits the widespread use of this model.

Even though the inclusion of structural information aided our lectin-glycan prediction model, we were somewhat surprised that, e.g., SASA information did not improve model performance even further, since most lectins had significant dependencies on these structural attributes (Figs. 3–4). Especially since a LectinOracle variant trained only on glycan-structure information (without access to monosaccharide identity) yielded a MSE of 0.591, which, if anything, presented a model that was slightly more performant than LectinOracle trained exclusively on sequences, indicating the information in structural data for this prediction task.

Yet, given the great success of AI models in the domain of protein structure prediction and foundation models[33,34]—purely trained on sequences—we reasoned that much of the structural information about glycan 3D conformation must already be contained in their sequence (similar to proteins[33]) and thus might be already used, and learned implicitly, by models such as LectinOracle. To test this, we set up an experiment to predict structural aspects of glycans purely from their sequence, via a herein trained AI model.

For this, we used a set-up similar to our traditional SweetNet model[32] to predict, purely from glycan sequence, structural attributes such as SASA or torsion angles. Since one glycan sequence usually had several conformers (and thus associated values), we adopted a different strategy here: instead of directly predicting angles (forcing the model to learn a weighted average of conformers), we predicted the parameters of a multimodal von Mises distribution (the circular analog of a Gaussian distribution[35]; Fig. 5c), which has recently shown great promise for similar tasks in protein-focused models[36]. The model predicted mixture weights, means, and concentration parameters for up to five components per angle, enabling representation of complex multimodal conformational landscapes. We could then use this model to (i) calculate a negative log-likelihood loss term of the observed torsion angles under the predicted von Mises distribution and (ii) sample points for predicted Ramachandran diagrams from this learned multimodal distribution (see Fig. 5e). This von Mises mixed model approach then natively supported multiple glycan conformers (trained on 6911 total conformers), since the model could simply assign separate probability density to the angle distribution for each conformer, and we are excited that this general model set-up could find more such applications in the future.

Supporting our hypothesis of sequence-inherent structure information, this von Mises model then readily outperformed strong baselines (the mono-/disaccharide-specific mean or median) and, e.g., resulted in angular RMSE prediction errors of merely ±7° (φ), ±9° (ψ), and ±10° (ω) for entirely unseen glycans (Fig. 5d). This performance was especially impressive given the inherent conformational flexibility of glycan conformers. Visualizing model predictions via Ramachandran plots then confirmed that realistic, and conformer-specific, torsion angle distributions have been learned by this von Mises-SweetNet-style model (Fig. 5e; Supplementary Fig. 15a), with the multimodal distributions capturing the probability density across different conformational states. This even included zero-shot problems such as torsion angles for disaccharides that have never been seen by our model (Supplementary Fig. 15b and c).

Further, we report excellent correlations between predicted and observed SASA values for monosaccharides in glycans (Fig. 5f; Pearson's $r = 0.94$), and moderate correlations with glycan flexibility (Supplementary Fig. 16). Overall, this makes us optimistic that more

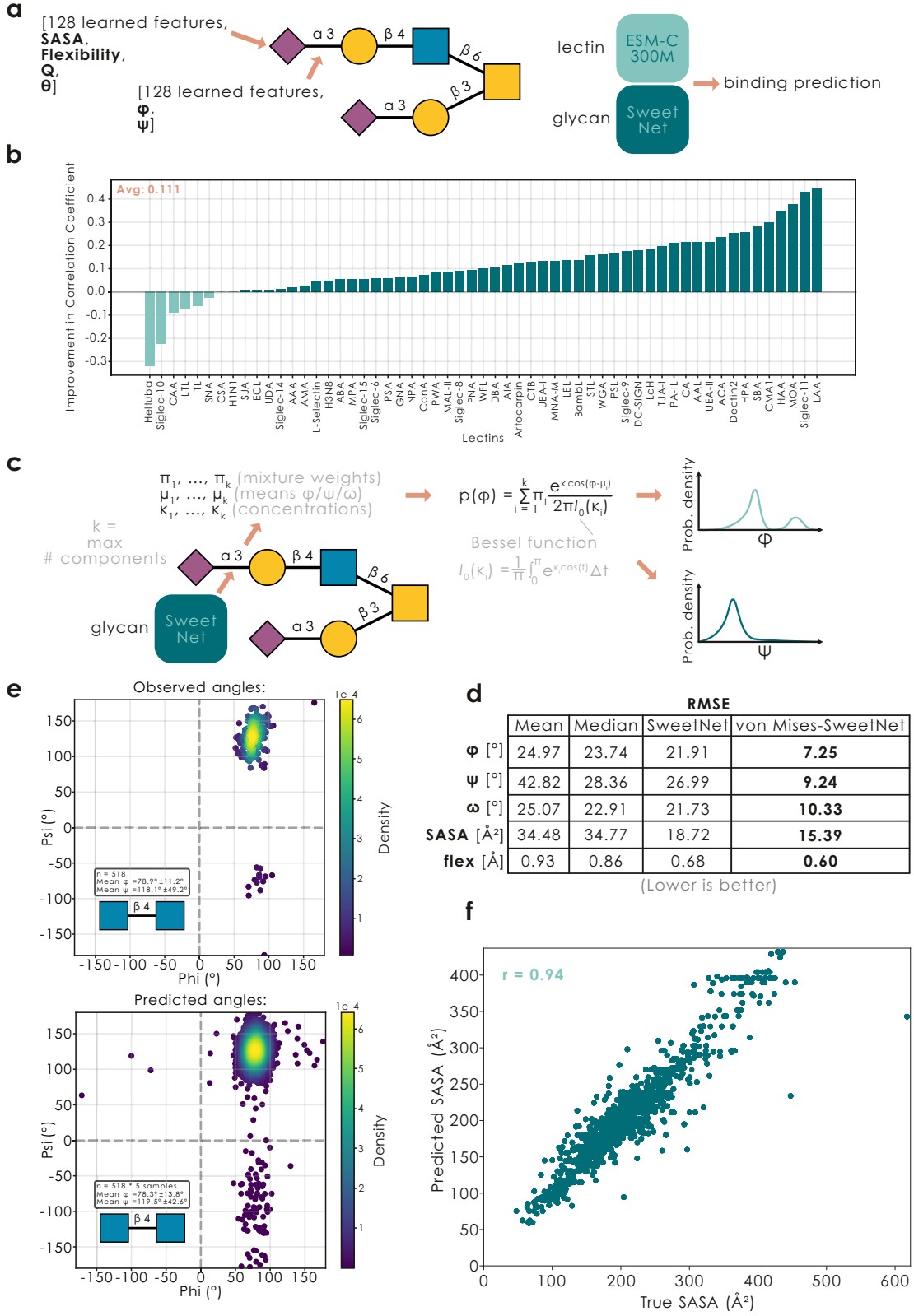

challenging structure prediction tasks can be attempted with glycans once more glycan structures become available, up to the eventual prediction of full 3D structural ensembles. Further, we stress that the prediction of torsion angle ranges for (i) never-before-seen disaccharides in (ii) arbitrary sequence contexts and (iii) in milliseconds is of immediate relevance for structural (glyco)biology as well as to generate structural features for a more general applicability of models

such as LectinOracle_struct, providing downstream tasks with structural glycan information for improved predictions.

## Discussion

Considering glycan structure when analyzing function is key. The aim of GlyContact is to make this domain of structural glycobiology more accessible to the broader research community, especially by bridging

**Fig. 5 | Glycan conformers are informative for AI models and can be predicted.**
**a** Glycan 3D attributes can be used by AI models operating on glycan-based graphs, with the example of a structure-informed version of LectinOracle.
**b** LectinOracle$_{struct}$ binding predictions correlate better with experimentally measured values. Shown via a bar graph is the improvement in Pearson's correlation coefficient of prediction vs experiment for the 58 well-known lectins analyzed above, comparing predictions by LectinOracle and LectinOracle$_{struct}$. **c** Schematic view of predicting torsion angles as multimodal von Mises distributions. Shown is the formula to calculate angle probabilities, given predicted parameters, by using the modified Bessel function of the first kind and order 0. **d** SweetNet-style glycan models can predict structural attributes from sequence alone. For the structural attributes of torsion angles (φ, ψ, ω), SASA, and flexibility, we provide

monosaccharide (SASA, flexibility) or disaccharide (torsion angles) specific baselines (mean, median), as well as model predictions for a model trained on only the most-frequent conformer ("SweetNet") or individual conformers ("von Mises-SweetNet"). Values are provided as RMSE (lower is better), with the best performance for each attribute bolded. **e** Ramachandran distribution of torsion angles for the disaccharide GlcNAcβ1-4GlcNAc as observed across all conformers ($n = 518$; drawn with the *glycontact.visualize.ramachandran_plot* function) or predicted by a von Mises-SweetNet-style model by sampling five points from the multimodal von Mises distribution created from predictions for each disaccharide sequence occurrence ($n = 518 \times 5$). **f** Predictions of monosaccharide-level SASA values from the test set, by a von Mises-SweetNet model, are strongly correlated with observed SASA values, quantified by the shown correlation coefficient as Pearson's r.

mature glycoinformatics routines on the sequence level with the spatial dimension. The integration with standard file formats and existing computational platforms, such as glycowork, ensures that GlyContact can be readily incorporated into existing research workflows.

We also show here that the conformational flexibility of glycans, including several conformer representatives, can be leveraged to understand glycan properties and functions such as lectin-glycan binding. Our analyses using GlyContact have revealed several important insights into glycan structure-function relationships: (i) Twin analyses can isolate the average effect of glycan motif addition on the overall structure (e.g., Fig. 2b, Supplementary Fig. 8), (ii) Structural properties of glycan motifs are sequence context-dependent, such as in different glycan classes (e.g., Fig. 2c, Supplementary Fig. 9), and (iii) Lectins have characteristic dependencies on a specific structural configuration of their binding motif(s) (e.g., Fig. 3, Fig. 4a, Supplementary Figs. 10–12). These insights have potential future implications for glycoengineering efforts and the rational design of glycan-targeting therapeutics.

Increasingly available glycan 3D structures will further extend these findings. Here, we caution that GlyContact does not remediate carbohydrate 3D structures. If the wrong glycan has been originally modeled in a PDB file, GlyContact will merely extract this structure, not correct it. For this, we point readers to dedicated software[12,37,38] and databases[39] for carbohydrate 3D remediation. Further, we caution users against generalizing from GlyContact-derived results using GlycoShape molecular dynamics structures to glycans in a glycoprotein context, as linker effects (influenced by the protein structure) will change structural properties of the attached glycan. We also note that, while the von Mises-SweetNet model accurately predicted individual torsion angles within an error of ±7–10°, and thus could generate realistic Ramachandran plots even for unseen disaccharides, we currently do not yet learn angle-angle dependencies and leave this next stepping-stone towards glycan 3D ensemble prediction to future work. Future approaches to this could, for instance, incorporate multivariate multimodal von Mises distributions (or copula-based models) to capture φ-ψ correlations, or employ conditional modeling where ψ predictions depend on predicted φ distributions within the same linkage. These avenues could be helped by the non-linear torsion-torsion correlation analyses we have implemented into GlyContact (Supplementary Fig. 4).

Next to analyzing additional molecular dynamics-derived structures, we also caution researchers that more structures from other sources, such as X-ray crystallography, will be required to solidify our understanding of trends and associations, as especially the quantitative distributions of conformational states still need to be experimentally validated. For this, however, advances in structural glycobiology will be necessary, as the poor electron density of extended glycans in this data type currently precludes such analyses on a routine basis. We note that current developments in protein structure prediction, with the release of AlphaFold3, have begun to offer users the option to also model glycan structures[40]. However, it has been pointed out that caution should be exercised as these glycan

structures then still are static snapshots and do not capture the broad conformational flexibility that glycans exhibit in molecular dynamics simulations[41]. We remain optimistic that future advances in this field will eventually yield more data to apply our herein presented methods to.

While analyzing lectin-glycan binding via weighted SASA values of the binding motif has resulted in clear and significant associations with glycan array-derived binding values here, we caution that even better results would be achieved if the SASA of the actually bound conformer would be used instead. However, this information is usually not available for analysis. We also note that the binding dependency on SASA/flexibility seems highly lectin-dependent and should be evaluated on a case-by-case basis. Lastly, analysis of flexibility is also context-dependent, as distance-based flexibility (as calculated here) assumes a relatively rigid reducing end, which would, e.g., not be a realistic assumption for free milk oligosaccharides. Especially for such cases, we recommend users to perform analyses with torsion-based flexibility, which does not suffer from the same limitation.

Our work here demonstrates the potential of combining glycan 3D information with quantitative binding to yield insights into lectin-glycan interactions. This is corroborated by related work[42], which has shown that informing the analysis of lectin-glycan 3D interactions with information from glycan arrays is a promising direction of research. Providing models such as LectinOracle with structural attributes such as monosaccharide-level flexibility or SASA has improved lectin-glycan binding predictions and future work should disentangle the marginal contribution of each structural attribute to this outcome. Similarly, while we here use a probe radius of 1.4 Å (representing a water molecule as the bulk solvent) to calculate SASA, other probe radii could be investigated to optimize this procedure. Future research should also evaluate other structural attributes, such as the spatial arrangement of hydroxyl groups in the binding motif, as to their potential to improve predictions.

Nevertheless, we also would like to note that the prediction tasks here, such as predicting structural attributes of glycans from their sequence alone, might become candidates for benchmark tasks for evaluating glyco-AI models[31], which could aid in identifying better ways to model glycans for applications in glycoinformatics[18]. Additionally, approaches building on our von Mises-SweetNet model might eventually find use cases in guiding the design of glycans with specific structural properties, such as in recent work on glycan hairpins[43].

Looking forward, GlyContact provides a foundation for emerging approaches in glycobiology and glycoinformatics. The standardized structural analyses enabled by GlyContact could also serve as modules within machine learning approaches in glycan structure prediction and functional applications[18]. We note here the important insight that we observe gains in the performance of our AI models when including glycan structural information that cannot be compensated for with a more expressive protein representation, cautioning against protein-only AI models that have become more and more widespread. As additional experimental methods for studying glycan structure emerge, GlyContact's modular design will integrate these data types.

We envision that community-driven development of our open-source software will continue to expand GlyContact's capabilities, establishing it as a central tool in structural (glyco)biology.

## Methods

### Data processing

For GlycoShape-derived structures, we retrieved all 717 glycans from GlycoShape that had accessible PDB files via the API, using the *glycontact.process.download_from_glycoshape* function (Supplementary Data 1). For all glycans, we retrieved all conformers (both alpha and beta versions of all cluster representatives), as well as the respective conformer frequencies. We then also calculated all their structural attributes (SASA, flexibility, ring puckering, polar angle, torsion angles) via the *glycontact.process.get_structure_graph* function and stored them in glycowork[14]-derived graphs in the *glycontact.process.glycan_graphs* module as a dictionary of form glycan sequence: structure-annotated glycowork graph.

For UniLectin-derived structures, we retrieved all structures from UniLectin3D that had a defined IUPAC-sequence of a co-crystallized glycan. This was then followed by manual inspection, and remediation, of the purported co-crystallized glycan and the actually present glycan in the PDB file. This process resulted in 1437 PDB structures containing at least one glycan that could be extracted (Supplementary Data 2). We then used the *glycontact.process.get_annotation* function to extract and process glycan coordinates from those files, followed by a recentering and relabeling of residue and chain numbers. All such processed structures were then stored in the *glycontact.process.unilectin_data* module, accessible as a dictionary of form glycan sequence: list of all processed PDB coordinates.

Glycan-binding data was sourced entirely from glycowork (v1.6), via the *glycowork.glycan_data.loader.glycan_binding* module.

GlyContact uses the Universal Input platform facilitated by glycowork (v1.6), allowing users to use any glycan nomenclature of their choice, which is automatically detected and converted. Further, we implemented a structure dispatcher, such that user-inputted glycans are first searched in GlycoShape-derived structures and, if unavailable, then searched amongst UniLectin3D-derived structures.

### Solvent-available surface area analysis

The solvent-accessible surface area (SASA) of glycan structures was calculated via the Shrake-Rupley algorithm as implemented in the MDTraj library[44], which uses a rolling sphere approach to estimate the surface area accessible to a solvent probe.

The SASA calculation was performed at the atom level using a probe radius of 1.4 Å (representing a water molecule as the bulk solvent), and then summed to obtain residue-level accessibility values. For monosaccharides with attached modifications (such as phosphocholine, sulfate, or acetyl groups), we attributed the SASA contribution of the modification to its parent residue to provide a complete picture of the modified monosaccharide's solvent exposure. As is standard practice in the study of biomolecular interactions, we hence used the solvent-accessible area as a proxy for availability for binders, both in terms of physical interaction and coordinated water molecules that can be freed for entropic gains upon binding. For the case of glycoprotein structures, this also included the SASA value of the linker amino acid.

When analyzing multiple conformations of the same glycan, we calculated weighted averages of SASA values, using the relative frequencies of different conformational clusters, to obtain a representative picture of the average solvent accessibility in solution.

### Flexibility analysis

We quantified glycan flexibility using two distinct approaches, as well as depending on the data source. For structures derived from X-ray crystallography, we calculated root-mean-square fluctuations (RMSF) from the temperature factors (B-factors) as distance-based flexibility using the equation:

$$\text{RMSF} = \sqrt{3 * \frac{B}{8 * \pi^2}} \tag{1}$$

where B represents the temperature factor for each atom. For the case of glycoprotein structures, this also included the flexibility of the linker amino acid.

For structures from the GlycoShape database, we measured inter-structure variability by calculating positional deviations across multiple conformers of the same glycan. We computed the mean absolute deviation (MAD) of atomic positions across conformers and converted this to an RMSF-equivalent measure as distance-based flexibility using a conversion factor of $\sqrt{(\pi/2)}$:

$$\text{RMSF} = \text{MAD} * \sqrt{\frac{\pi}{2}} \tag{2}$$

Lastly, to avoid alignment artifacts near the reducing end, we also assessed torsion-based flexibility for each residue by averaging the differences between torsion angles (φ, ψ, and, if applicable, ω) for each linkage in which a residue was involved. For each glycosidic linkage across all available conformers, we calculated the circular standard deviation of φ, ψ, and ω angles separately, then averaged these three values to obtain a single flexibility measure per linkage. For residues participating in multiple linkages, we averaged the flexibility values across all their associated linkages to derive a final per-residue torsion flexibility score expressed in degrees, where higher values indicate greater angular variability across conformers.

For glycans with multiple conformers, we calculated the above as a weighted average of flexibilities, informed by relative frequencies of different conformational clusters. For distance-based flexibility, values were calculated at both the atomic and residue levels, with residue-level values determined by averaging the flexibility metrics of all atoms within each monosaccharide.

### Glycosidic torsion angle calculation

Glycosidic torsion angles were calculated for each glycosidic linkage, including phi (φ), psi (ψ), and, where applicable, omega (ω) angles. The φ angle was defined by the torsion between O5-C1-O$x$-C$x$ atoms (where $x$ represented the position number of the glycosidic linkage), describing the rotation around the C1-O$x$ bond. For sialic acid residues, we used O6-C2-O$x$-C$x$ atoms instead. The ψ angle was defined by the torsion between C1-O$x$-C$x$-C$y$ atoms (where $y$ represents the next carbon in the ring sequence), characterizing the rotation around the O$x$-C$x$ bond.

For 1/2-6 linkages specifically, we also calculated the ω angle, defined by the torsion between O6-C6-C5-O5 atoms, which describes the rotation around the exocyclic C6-C5 bond. This additional torsion angle is critical for accurately representing the flexibility of 1–6 linkages, which possess an extra degree of freedom compared to other glycosidic connections.

All torsion angles were calculated using the standard method for determining dihedral angles from four sets of 3D coordinates. For the φ angle, the calculation was performed using the following equation:

$$\varphi = \text{atan2}\left(\left[(n_1 \times n_2) \cdot \frac{v_2}{|v_2|}\right], n_1 \cdot n_2\right) \tag{3}$$

where $v_1$ = C1 - O5, $v_2$ = O$x$ - C1, and $v_3$ = C$x$ - O$x$ are the bond vectors, and $n_1 = v_1 \times v_2$ and $n_2 = v_2 \times v_3$ are the normal vectors to the planes. The symbols × and · represent the cross product and dot product, respectively. We used the four-quadrant inverse tangent function (atan2) to ensure the correct sign of the angle. For the case of glycans

covalently linked to proteins (i.e., glycoproteins), we also report the torsion angles (φ/ψ) between the reducing monosaccharide and the amino acid to which the glycan is attached. Angular values are reported in degrees, rounded to two decimal places.

## Monosaccharide conformation analysis

Ring conformations of monosaccharides were analyzed using the Cremer-Pople puckering parameters[45]. For pyranose rings, we analyzed atoms C1-C5 and O5. For furanose rings (e.g., Araf, Galf, or Fruf), we used C1-C4 and O4. For sialic acids, we analyzed the 6-membered ring formed by C2-C6 and O6.

We first determined the geometric center of the ring atoms and defined a reference plane. We then calculated the normal vector to this plane and projected the atoms onto it. The deviation of each atom from this plane ($z_\diamond$) was used to calculate puckering amplitudes ($q_m$) and phase angles ($\varphi_m$) using the following equations:

$$q_m \cdot \cos(q_m) = \frac{2}{n} \cdot \sum_j z_j \cdot \cos\left(\frac{2\pi m j}{n}\right) \tag{4}$$

$$q_m \cdot \sin(q_m) = \frac{2}{n} \cdot \sum_j z_j \cdot \sin\left(\frac{2\pi m j}{n}\right) \tag{5}$$

where $n$ is the number of ring atoms, $j$ is the atom index (0 to $n$-1), and $m$ ranges from 1 to $n/2$. The total puckering amplitude Q was calculated as the square root of the sum of squared individual amplitudes.

For 6-membered pyranose rings, we used the polar coordinate representation with amplitude $q_2$, $q_3$, and phase angle $\theta = \tan^{-1}(q_2/q_3)$. Conformations were then classified as chair ($^4C_1$ or $^1C_4$, with $\theta < 45°$ or $\theta > 135°$, respectively, with inversions for L-monosaccharides), boat, or skew-boat forms. For boat conformations ($B_{1,4}$, $B_{2,5}$, $B_{3,6}$), we analyzed the $\varphi_2$ angle at multiples of 60°, while skew-boat conformations ($^1S_3$, $^2S_6$, $^3S_1$, etc.) were identified at 30° off these values.

For 5-membered furanose rings, we used $q_2$ and $\varphi_2$ to classify conformations as either envelope (E) or twist (T) forms. Envelope conformations (C3-endo, C4-endo, O4-endo, C1-endo, C2-endo) were assigned at phase angles corresponding to multiples of 72°, while twist conformations ($^3T_4$, $^4TO$, $OT_1$, etc.) were assigned at 36° off these values.

For sialic acids, we used a modified approach accounting for their distinct structure, classifying them primarily as $^2C_5$ chair, $^5C_2$ chair, or boat/skew forms based on the calculated θ angle and the relevant phase angles.

## Glycan structure alignment

We implemented a structural alignment procedure using the SVD-based Kabsch algorithm that finds the optimal rigid transformation to minimize distances between corresponding atoms in two glycan structures. Atomic coordinates were extracted from PDB files, using either specified residue IDs or the entire glycan structure. For main chain-only alignments, we selected atoms C1-C5 and O5; for complete structural comparison, all non-hydrogen atoms were included.

For fast alignment, the algorithm first centers both coordinate sets by subtracting their respective centroids. A $k$-d tree spatial indexing structure identifies the closest corresponding atoms between the reference and mobile structures. The covariance matrix between matched points is computed, followed by singular value decomposition (SVD) to determine the optimal rotation matrix, with special handling for reflection cases. The final transformation applies both rotation and translation to align the mobile structure with the reference.

The root-mean-square deviation (RMSD) between structures after superposition was calculated as:

$$RMSD = \sqrt{\frac{1}{n} \cdot \sum (|p_i - q_i|)^2} \tag{6}$$

where $p_i$ represents each point in the transformed mobile structure and $q_i$ represents its corresponding closest point in the reference structure. Users also have the option to replace this with Nelder-Mead optimization for a slower, but typically more accurate, alignment.

When comparing glycans with multiple conformers, alignments were performed between all pairs of conformers, and the pair with the lowest RMSD was selected as the best structural match.

## Lectin binding motif analysis

As for predicting lectin-glycan interactions below, we first formed the intersection of glycans for which we had both (i) glycan array-derived binding data stored in glycowork and (ii) structural data from GlycoShape. Using lectin binding motif annotations from the literature[24] and glycowork[14], including positional specifiers such as whether a motif had to be terminal, we then gathered binding data and SASA/flexibility for all those glycans that exhibited a binding motif. For each such glycan, we had to aggregate monosaccharide-level SASA and flexibility values within a motif and across motifs. Extensive testing revealed averaging at both levels to yield the highest correlation coefficients with binding, on average. Next, linear least-squares regression, followed by calculating Pearson's correlation coefficient, between SASA/flexibility and z-score transformed binding data allowed us to probe the sensitivity of each lectin to the structural properties of its binding motif(s).

## Predicting lectin-glycan interactions

After forming the intersection of glycans for which we had both (i) glycan array-derived binding data stored in glycowork and (ii) structural data from GlycoShape, we split our data 90% and 10% into training/validation sets on the protein level and trained three deep learning models to predict z-score transformed binding data, given a protein sequence and a glycan sequence as input: (i) a typical LectinOracle model, as described previously[30], using the graph convolutional neural network SweetNet[32] as the glycan encoder, (ii) LectinOracle_struct, which also used a SweetNet architecture yet additionally processed structural attributes (SASA, flexibility, ring puckering, polar angle, torsion angles), derived from the *glycontact.process.get_structure_graph* function, and (iii) a LectinOracle model that only used structural attributes (and no monosaccharide embeddings). All models used the ESMC-300M model[46] as a protein encoder, and used a fully connected neural network for the final prediction. For all training, we used MSE loss, AdamW as an optimizer, a batch size of 128, and a learning rate of 0.0005 (reduced by 80% upon a plateau of four epochs without validation loss improvement). Models were trained for up to 150 epochs, with an Early Stopping criterion of 20 epochs without validation loss improvement. Models were trained using PyTorch (v2.5.1) and PyTorch Geometric (v2.6.0) on a single NVIDIA GeForce RTX 4060 GPU.

## Predicting structural attributes from glycan sequences

We used the *glycontact.process.get_structure_graph* function to calculate SASA, flexibility, and torsion angles (φ, ψ, ω) for all conformers of all GlycoShape-derived glycans. Each glycan was represented as a graph where monosaccharide nodes contained SASA and flexibility values, and linkage nodes contained torsion angles. Node features were encoded using a learned embedding layer (dimension 128) based on string labels, while edges were constructed bidirectionally between all connected residues. Data were split into training (70%) and validation (30%) sets on the sequence level using DataSAIL[47], ensuring that

no conformers of the same glycan were present in both sets. Baselines were calculated as the monosaccharide-specific (SASA, flexibility) or disaccharide-specific (torsion angles) mean/median values.

We developed a von Mises-SweetNet model to simultaneously predict multimodal von Mises distribution parameters for torsion angles and continuous values for SASA/flexibility. The architecture consisted of three Graph Isomorphism Network (GIN[48]) layers, each containing two linear transformations (128 hidden units), ReLU activation, batch normalization, and 30% dropout. The model employed separate prediction heads: one for von Mises parameters (φ, ψ, ω angles) and another for SASA/flexibility regression. The von Mises head used a two-layer network (128 → 64 → 5 units) to predict mixture weights, means, and concentration parameters (κ) for up to five components per angle, representing the multimodal nature of torsion angle distributions across conformers.

For the von Mises mixture model, means were constrained to ±180° using tanh activation, while concentration parameters were generated via softplus transformation with clamping ($\kappa \leq 10$) to prevent numerical instability. Mixture weights were normalized using softmax. The negative log-likelihood loss for angle prediction was calculated as $L = -\log(\Sigma_i\, w_i \times VMF(\theta \mid \mu_i, \kappa_i))$, where $w_i$, $\mu_i$, and $\kappa_i$ are the weight, mean, and concentration of component i, and VMF is the von Mises probability density function. SASA and flexibility used mean squared error loss. Node masking ensured that only linkage nodes contributed to angle loss and only monosaccharide nodes to SASA/flexibility loss.

A comparison GIN-SweetNet model was formulated to predict torsion angles as discrete values using MSE loss, trained only on the most frequent conformer per glycan. In contrast, the von Mises model was trained on all conformers ($n = 6911$ total), to capture the full conformational distribution. Models were trained using the Adam optimizer and a linear warm-up cosine annealing for the learning rate schedule. Models were trained using PyTorch (v2.5.1) and PyTorch Geometric (v2.6.0) on a single NVIDIA GeForce RTX 4060 GPU.

For trained models, SASA and flexibility prediction performance was evaluated as RMSE (weighted by conformer frequency), whereas torsion angles were evaluated as angular RMSE (GIN-SweetNet) or angular RMSE of the best-matching component (von Mises-SweetNet). Ramachandran plots from trained models were generated by predicting mixture weights, means, and concentrations for all glycans in the test set that contained a given disaccharide and then sampling from the predicted von Mises distribution of only the nodes representing the disaccharide of interest.

### Statistical analysis
All statistical testing has been done in Python 3.12.6 using the glycowork package (version 1.6), the scipy package (version 1.11), and the GlyContact package (version 0.3.0). Testing differences between two groups used paired t-tests in the case of paired data and Mann-Whitney U-tests for unpaired data. Multiple testing correction was performed via the Benjamini-Hochberg procedure. Effect sizes are reported for paired data as Cohen's $d_z$, calculated via the glycowork implementation (*glycowork.glycan_data.stats.cohen_d*; version 1.6).

### Reporting summary
Further information on research design is available in the Nature Portfolio Reporting Summary linked to this article.

## Data availability
Unless otherwise stated, all data supporting the results of this study can be found in the article, supplementary, and source data files. Stored datasets from UniLectin3D can be found within glycontact [https://github.com/lthomes/glycontact][49]. The molecular dynamics data analyzed here are available at GlycoShape. Source data are provided with this paper.

## Code availability
All used code can be found at glycontact [https://github.com/lthomes/glycontact][49].

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

## Acknowledgements

The authors would like to thank Silvia D'Andrea, Ojas Singh, Beatrice Tropea, and Elisa Fadda for simulating glycan 3D conformations and making them available to the community via GlycoShape. This work was supported by a Branco Weiss Fellowship – Society in Science awarded to D.B.; by the Knut and Alice Wallenberg Foundation awarded to D.B.; the Hasselblad Foundation awarded to D.B.; and the University of Gothenburg, Sweden, awarded to D.B. The funders had no role in study design, data collection and analysis, decision to publish or preparation of the manuscript.

## Author contributions

D.B. and L.T. conceptualization; Z.A., D.B., R.J., and L.T., formal analysis; D.B., resources; D.B. and L.T., data curation; D.B. and L.T. writing–original draft; Z.A., D.B., R.J., and L.T., writing–review & editing; D.B. visualization; D.B., supervision; D.B., funding acquisition; Z.A., D.B., R.J., and L.T., methodology; Z.A., D.B., R.J., and L.T., validation.

## Funding

## Competing interests

D.B. is consulting on glycobiology-related topics via SweetSense Analytics AB. The remaining authors declare no competing interests.
