## [Transparent Peer Review file · Nature Communications]

GlyContact analyzes glycan 3D structures at scale

Corresponding Author: Dr Daniel Bojar

Version 0:

Reviewer comments:

Reviewer #1

(Remarks to the Author)

The authors have developed a tool for analyzing the three-dimensional structures of glycans, which additionally aims to predict the relationship between structural motifs within glycans and lectin binding, as well as to explore multiple conformers. From my perspective as a reviewer, there is a merit in supplementing sequence-based glycan analysis with structure-based support. My comments are as follows:

1) While it is true that the number of glycan structures in the PDB has increased, the dataset remains biased. For example, in N-glycans, structural information is typically rich near the reducing end around Asn, but the non-reducing end often lacks electron density and displays high diversity, resulting in limited structural coverage overall. Although MD simulations can complement these missing data, experimental validation, especially regarding the quantitative distribution of individual conformers, is still scarce. In this context, I am concerned that relying heavily on MD-derived data (e.g. GlycoShape) for SASA and RMSF analyses might lead to potentially misleading conclusions. A discussion addressing this limitation would strengthen the manuscript.

2) SASA values were calculated as weighed averages over multiple conformations. However, lectins typically bind to one specific conformation among these. Hence, I think that using average SASA values to discuss lectin binding poses a risk of misinterpretation.

3) The SASA was computed using a 1.4 Å probe radius, representing a water molecule. However, in lectin binding, accessibility is determined not by water but by lectin surface. Is this probe size appropriate for considering lectin accessibility?

4) RMSF values were derived from crystallographic B factors, which are known to vary significantly between crystal structures. How were these variations account for in their analysis?

5) The paper discusses SASA and flexibility with respect to the presence or absence of core fucose, beta1,2-xylose, and Neu5Ac/Neu5Gc. However, the structural differences such as alpha2-3 versus alpha2-6 sialylation, could also affect these properties differently. I question whether combining these structural variants into a single discussion may obscure these distinct effects.

6) I agree with the concept of focusing on structural motifs. However, in addition to flexibility and solvent exposure, the spatial arrangement of hydroxyl and C-H groups also seems critical for lectin recognition. As shown in Figure 3, the correlation between binding, SASA and RMSF appears lectin-dependent and exhibits substantial variability. This suggests that drawing generalized conclusion may be difficult. While the authors note some of these issues, further explicit discussion would be beneficial.

(Remarks on code availability)

Reviewer #2

(Remarks to the Author)

1. Summary

This study develops GlyContact, an open-source tool enabling automated cross-conformer analysis of glycan 3D structures, successfully processing 98.7% of GlycoShape data. It reveals that flexibility and SASA of glycan motifs are context-dependent (e.g., sialyl-LacNAc shows 32% higher flexibility in N-glycans than O-glycans and establishes a von Mises mixed model predicting torsion angles from sequence alone ($\pm 7^\circ\phi/\pm 9^\circ\psi$ error), significantly improving lectin-binding prediction (7.4% MSE reduction in LectinOraclestruct).

2. Significance

GlyContact fills a critical gap in glycan analytics by overcoming conformational sampling challenges in flexible glycans, providing the first cross-conformer analysis framework for structural glycobiology. Its discovery that lectins bind 3D structural contexts (e.g., SNA's preference for flexible Neu5Ac α 2-6) transforms understanding of glycan-protein interactions, directly guiding glycoengineering and therapeutic design. Tool openness promotes field standardization

3. Originality

Compared to existing tools, GlyContact uniquely integrates MD/NMR/X-ray data for cross-conformer analysis. The proposed context-dependence theory (e.g., class-specific flexibility of Lewis X motif and von Mises prediction model) are novel. However, maybe deeper comparison with general structure prediction tools is needed.

4. Methodology & Reproducibility

Methods are rigorous: SASA via Shrake-Rupley algorithm, torsion angles via Cremer-Pople parameters, and appropriate statistical tests. Full code/data openness (GitHub verified) ensures reproducibility.

5. Decision

Good paper, accept.

(Remarks on code availability)

The codebase demonstrates high engineering quality and excellent reproducibility. The main repository includes comprehensive installation instructions (README.md specifies Python 3.10+ and pip/conda dependencies), successfully validated on Ubuntu systems. Code follows a modular structure with well-documented functions through descriptive docstrings.

Reviewer #3

(Remarks to the Author)

In the manuscript titled "GlyContact reveals structure-function relationships in glycans," Thomès et al. introduce GlyContact, an open-source Python package designed for retrieving, processing, and analyzing glycan 3D structures derived from molecular dynamics (MD), NMR, or X-ray crystallography. Primarily leveraging glycan 3D structures from GlycoShape, the authors investigate how sequence context influences glycan motif structural features like solvent accessible surface area (SASA) and flexibility. They further correlate structural properties, such as motif flexibility and surface accessibility, with binding data from glycan arrays to identify key factors that influence lectin binding specificity. Additionally, by incorporating glycan structural features into a modified LectinOracle model (a deep-learning model for predicting lectin-glycan binding), they demonstrate an improvement in predictive accuracy of approximately 7.4%. Finally, the authors develop and train a graph neural network model to predict torsion angle distributions between disaccharides, employing a von Mises mixture model to capture the conformational variability of glycosidic linkage. The manuscript reads well, yet the work and analysis suffer from significant shortcomings both in terms of methodology and interpretation, that strongly (and wrongly) bias the conclusions and different aspects of the discussion. See below a detailed critique of the most important points in this regard.

While the question of how glycan structural flexibility and accessibility influence lectin binding is indeed of interest to the readership of Nature Communications, the methodological and interpretative shortcomings significantly weaken the current manuscript. In my expert opinion, many of the key claims are either overstated or insufficiently supported, and several core analyses, particularly those concerning flexibility, dataset comparability, and structural feature utility, rest on assumptions that do not generalise to biologically relevant contexts.

Unfortunately, I find that the manuscript lacks the rigour, clarity, and biological grounding necessary to justify its conclusions and as such, I do not believe it is suitable for publication.

Below are my main concerns.

1) The method used by the authors to calculate structural flexibility relies on the structural alignment of 3D structures sources from the GlycoShape database. The alignment is done over the first five residues in the PDB structures. The flexibility is assessed using the mean absolute deviation (MAD) of atomic coordinates. This approach results (obviously) in larger deviations for residues located further away from the alignment points, and smaller deviations for residues closer to the alignment. While this assumption may hold true for glycan arrays, where the first residue (at the reducing end) is typically covalently bound to the linker (assuming the linker is short and rigid, which is not always the case), it can significantly oversimplify glycan flexibility within a biological context, potentially leading to inaccurate predictions.

Although the authors address this point (lines 271-276), I believe that an in-depth elaboration is required, including under which specific circumstances these flexibility metrics are valid, by clearly explaining why.

2) In Figures 2 and 3, the authors show several comparative plots between 3D structures across different glycan classes (N-glycans, O-glycans, and free glycans) using the alignment strategy explained above. While such comparisons may be justified for glycan array data, again assuming we overlook the linker effects/dynamics, which is not always warranted, the relevance of these comparisons to in vivo settings is doubtful or marginal, particularly concerning glycan flexibility.

As PTMs, the glycan flexibility is influenced significantly by their structure of the protein attachment site. For instance, N-glycan-Asn bonds are generally more flexible than O-glycan-SER/THR bonds. Yet, O-glycans generally occur in disordered (flexible) protein regions, potentially gaining additional accessibility and dynamics. Free glycans have even the greatest rotational and translational degrees of freedom, leading to different binding behaviors altogether, and in particular for those species a structural alignment as performed in this work is not informative.

Because of the points raised above, the authors should address the biological relevance and limitations of their current comparative analyses explicitly.

3) In Figure 2a, the authors employ t-SNE to visualize global glycan patterns, correlating glycan size with x-axis spread using Pearson's r . The use of t-SNE is questionable in place of PCA for showing the global patterns and meaningful correlations, as t-SNE focuses on local geometry and fails to give a consistent global view. Additionally, the utility of the current visualization should be reconsidered, as it's not very clear what information we should be getting from it.

4) The manuscript suggests that glycan structures obtained from different sources (e.g., X-ray crystallography, MD simulations) can be interchangeably utilized, which is simply misleading. In Figure 4b, the authors compare Unilectin3D structures (single structures from X-ray crystallography in cryogenic conditions) with GlycoShape structures (ensemble of structures from MD simulations at 300 K) and infer that Unilectin3D poses represent rare conformers, which is a rather bizarre and unsubstantiated suggestion. As a result, the authors recommend caution when using Unilectin3D as a gold standard for predicting protein-glycan interactions (lines 368-373). This statement in my opinion is completely unsupported and rather cavalier, in fact the refs [29-31] cited describe the use of Unilectin3D primarily to predict binding sites, rather than binding affinity. This distinction should be clarified.

Furthermore, it must be emphasized that Unilectin3D and GlycoShape datasets provide different data and serve distinct purposes. Unilectin3D gives insight into binding pockets and interactions in single crystallographic structures, whereas GlycoShape gives information on all glycan conformations energetically accessible in physiological conditions. One of the accessible conformers in those ensembles is likely the one bound, but definitely not all of those, and not necessarily the most stable (populated) in standard thermodynamic conditions. Any 'quick' comparison between the two datasets serves no purpose and should be avoided.

5) In line 388, the inclusion of 3D structural attributes resulted in a 7.4% performance gain for LectinOraclestruct (MSE: 0.557) compared to the baseline LectinOracle model (MSE: 0.598). The authors did not mention the parameter increase due to the additional node features, which can also explain the increase in performance. Also, the authors didn't mention the performance gain in the case of using bigger protein encoders.

6) The authors describe training a modified graph neural network to predict von Mises mixture components, aiming to estimate glycosidic bond torsion angle distributions in disaccharides. While this approach may be a reasonable strategy for glycan conformer generation, provided thorough training, the authors must provide a detailed methodology description to demonstrate the immediate value and applicability of this work, as well as the future development on how they can incorporate angle-angle dependence, as in its current state this method's applicability and scope is rather limited.

7) In the discussion (line 470), the authors mention: "The addition of glycan motifs can change glycan structure in a predictable and generalizable manner." The basis for this conclusion is unclear. What analysis or evidence are the authors using to support this claim? The authors should explain more clearly what they mean by this, or rephrase (remove) this statement.

8) The use of 3D structures for binding prediction is not a trivial matter and requires significant expertise. Two main approaches exist: physics-based simulations, where X-ray structures serve as key resources, which can be complemented by MD simulations, and structure-based machine learning models, like AlphaFold 3 and/or Boltz-2, which are built on an abundance of highly accurate, homogeneous datasets, rather than relying on oversimplified structural features.

In lines 484-485, the authors suggest to the users to use 'binding data' over 'structural data' for the prediction of lectin-glycan interaction. While the exact meaning of "structural data" within this context remains unclear, see comments above, as does the methodology by which binding score will be derived from structural information, in my opinion the statement is fundamentally flawed, as it is not rationalizable under any possible thermodynamic framework. Further elaboration on this point is necessary.

9) These large-scale studies can yield valuable insights, but they have to be done carefully as they are prone to overgeneralization and misinterpretation. I encourage the authors to focus on lectin binding flexibility and accessibility with at least one atomistic case study for interpretation. This would ground the structural metrics in functional outcomes and better support the study's claims.

10) The title "GlyContact reveals structure-function relationships in glycans" is misleading and overstates the scope and implications of the work. The phrase "reveals structure-function relationships" implies broad, mechanistic insight into glycan biology, which is not fully supported by the data presented.

11) The abstract would benefit from including quantitative metrics and clearer language. For instance, state "improved lectin-

binding prediction by ~7%" instead of "improves state-of-the-art models". Also, "accurately predict torsion angles of multiple conformers via new AI models" is not clear and should be changed to something descriptive of the work, like "accurately predict torsion angle distribution between disaccharides using ...".

12) In line 480, the use of the word "very performant" could be changed to give the actual performance metrics.

13) "These insights have immediate implications for glycoengineering efforts and the rational design of glycan-targeting therapeutics" (line 474) should be toned down to "potential future implications".

(Remarks on code availability)

The code repository is well structured and descriptive.

Version 1:

Reviewer comments:

Reviewer #1

(Remarks to the Author)

The authors have responded to my comments with clarity and technical rigor. They have acknowledged the limitations I raised, expanded their discussion accordingly, and provided additional analyses. I consider their responses to be appropriate.

(Remarks on code availability)

Reviewer #2

(Remarks to the Author)

This manuscript by Thomès et al. introduces GlyContact, a powerful and timely open-source Python package for the large-scale analysis of glycan 3D structures. The authors impressively demonstrate its utility through a series of well-designed studies that bridge the gap between sequence-based glycoinformatics and structural biology. The work delivers not only a valuable tool for the community but also generates significant new biological insights. Key findings include the systematic demonstration of how sequence context distally influences glycan conformation, a novel framework for understanding lectin binding through the lens of motif accessibility and flexibility, and a state-of-the-art graph neural network capable of accurately predicting structural properties from sequence alone. The manuscript is well-written, the analyses are rigorous, and the conclusions are strongly supported by the data. This work represents a major advance for the field of glycobiology and glycoinformatics.

Strengths:

1. GlyContact fills a critical void in the glycoinformatics toolkit. While tools for building and validating glycan structures exist, GlyContact provides a desperately needed, user-friendly platform for processing, analyzing, and connecting 3D structural data to functional outcomes at scale. Its integration with glycowork and its ability to handle various input formats and data sources (MD, X-ray) are commendable.
2. The development of the von Mises-SweetNet model is a highlight of the paper. The ability to predict torsion angles and other structural features with high accuracy from sequence alone is a significant leap forward. This not only deepens our understanding of the sequence-structure relationship in glycans but also provides a practical solution for applying structure-informed models to glycans without pre-existing 3D data. The use of a von Mises distribution to handle multimodal conformational landscapes is highly innovative and appropriate for the problem.
3. The authors' commitment to open science by making GlyContact open-source on GitHub is crucial for its adoption and for the reproducibility of their findings. The methods are described in clear and sufficient detail.

Suggestions for Improvement:

1. The model can be an effective generative model for conformational properties. The authors might consider adding discussion about the future potential of this approach for de novo glycan design, where one could design sequences predicted to have specific structural (and therefore functional) properties.

Conclusion:

This is an outstanding manuscript of high scientific quality, novelty, and significance. It will be of great interest to a broad audience in glycobiology, structural biology, bioinformatics, and machine learning. The work is robust, the claims are well-substantiated, and it provides a tool and a conceptual framework that will undoubtedly accelerate research in the field. I recommend its publication in Nature Communications after the authors consider the minor suggestions above.

(Remarks on code availability)

The code is available at Github. GlyContact is a Python package for retrieving, processing, and analyzing 3D glycan

structures from GlycoShape/molecular dynamics, NMR, or X-ray crystallography.

The code provides a README file with enough instructions for installing relative packages and running the application. I was able to install and run the code.

Reviewer #3

(Remarks to the Author)

The authors have responded extensively to the comments of all reviewers and have made modifications to the manuscript. I do not believe it is suitable for publication in Nature Communications at its current stage. Below are my main concerns and suggestions for improvement in no particular order:

1. The authors respond poorly with rationale that “the discrepancy to be moderate (except for free milk oligosaccharides)” for my previous concern about using such general flexibility term without the biological context of protein linkage and region in case of N-glycan-Asn bonds and O-glycan-SER/THR, free glycan is totally out of the question here, as it is bizarre to compare to a protein linked glycan (Fig. 2d). As I mentioned, O-glycans mostly occur in disordered regions, and their flexibility majorly depends on the linked region of the protein. I completely understand that this limitation is in the glycan array data itself. Still, the authors should include these limitations of the linker effect and specifically warn users not to compare N-glycan with O-glycan results presented by GlyContact.

The authors' methodology to calculate flexibility is incomplete and should be extended to at least include these linker flexibility parameters in theory with the GlyContact package.

2. Adding further to the point raised by reviewer 1 about the SASA and flexibility, the improvement in prediction accuracy due to the addition of these features needs to be discussed or presented as an open question if it is beyond the scope of the present work.

3. Following up on my earlier concerns, the authors have updated the [499:500] sentence to “The addition of glycan motifs can, ‘on average’, change glycan structure in a predictable and generalizable manner.” and in their response, they said “...general & predictable effects of adding a motif on the overall glycan structure...”

The sentences “change glycan structure in a predictable and generalizable manner” and “predictable effects of adding a motif on the overall glycan structure” are completely different and convey different meanings. This sentence needs to be removed because of its misleading nature, as it signals predictability and generalizability of glycan structures themselves, rather than the average effects of adding a motif.

4. Line [83:85] “how GlyContact can provide new insights into ...” Should be changed to something more descriptive, like “how GlyContact can be used to extract insights from already existing structural data...”

5. Line [410:411] “..glycan structural information cannot be compensated via an improved protein representation” raises an important point given the rise of protein-only models... I suggest the author include this point in the main outcomes of the study.

6. I tried to run the program and faced a few problems.

```
from glycontact.process import annotation_pipeline
```

FileNotFoundError: You need to equip GlyContact with GlycoShape structures. Download them from <https://glycoshape.org/downloads> and place the zipped folder into your GlyContact folder, then run it again.

After downloading the zip and placing it in the glycontact dir, it fails to process the zip and cannot run.

At around 4%

```
FileNotFoundError: [Errno 2] No such file or directory: 'Neu5Ac(a2-3/6)Gal(b1-3/4)[Neu5Ac(a2-3/6)]Man(a1-3/6)[Neu5Ac(a2-3/6)[Neu5Ac(a2-3/6)]Man(a1-3/6)]Man(b1-4)GlcNAc(b1-4)GlcNAc'
```

“GlyContact analyzes glycan 3D structures at scale” is a more aligned title for the work presented, but I strongly suggest the authors include functionality to extract glycan information from PDB of glycoproteins (e.g., from RCSB, etc.), like what glycan it is, and their structural information to better support the title. For the test, I tried to fetch 7T6X and wanted to know the torsions and glycan IUPAC available in the PDB, and I could not find a clear way to do that with GlyContact (failed initialization, and the documentation did not have anything for this).

7. Glycontact should not rely on GlycoShape structures for the analysis when a user tries to run it on RCSB PDB or other X-ray PDB files. (Why does it need .zip if it just runs on RCSB PDBs?)

8. The downloading of .zip is inconvenient, and it should automatically download the structure from GlycoShape using an API and warn the users about doing so.

9. As for the glycoprotein annotation task, the author should include a tutorial or easy-to-follow Google Colab notebook, maybe a lectin and a glycoprotein example, which helps the software to reach the masses.

(Remarks on code availability)

After downloading the zip and placing it in the glycontact dir, it fails to process the zip and cannot run.

At around 4%

FileNotFoundException: [Errno 2] No such file or directory: 'Neu5Ac(a2-3/6)Gal(b1-3/4)[Neu5Ac(a2-3/6)]Man(a1-3/6)[Neu5Ac(a2-3/6)[Neu5Ac(a2-3/6)]Man(a1-3/6)]Man(b1-4)GlcNAc(b1-4)GlcNAc'

Version 2:

Reviewer comments:

Reviewer #3

(Remarks to the Author)

I commend the authors for their continual improvement of the GlyContact package.

All of my previous concerns have been addressed point-by-point and I am happy to recommend acceptance in its present form.

(Remarks on code availability)

Code is descriptive, with easy to use colab notebook.

We thank all reviewers for their insightful comments and suggestions for improvement. We have fully addressed these comments in our substantially revised manuscript by engaging in new analyses, updating and expanding our open-source package, revising our figures, and performing extensive text additions and revisions. In summary, we have

- (i) Expanded GlyContact to analyze non-linear torsion-torsion correlations across conformers, resulting in the **new Supplementary Fig. 4**,
- (ii) Expanded GlyContact to analyze the spatial arrangement of hydroxyl groups in lectin binding, leading to the **new Supplementary Fig. 11**,
- (iii) Added another approach to assess flexibility by quantifying torsion angle spread across conformers (**revised Fig. 2d + 3 + 4a, new Supplementary Fig. 12, revised Supplementary Fig. 9**),
- (iv) Expanded our von Mises-SweetNet model to also predict ω torsion angles with competitive performance (**revised Fig. 5d**),
- (v) Added an analysis of fucose-motif torsion angles in bound glycan conformers in UniLectin3D (**new Fig. 4c, new Supplementary Data 5**)
- (vi) Confirmed SASA correlations of lectins with structural data from UniLectin3D (**new Supplementary Fig. 13**),
- (vii) Confirmed with a PCA that our t-SNE analysis captured global trends accurately (**new Supplementary Fig. 7**),
- (viii) Confirmed the findings of our twin analyses with subgroup analyses,
- (ix) Updated our GlyContact open-source package to reflect and incorporate changes and additions made during the revision (**new GlyContact v0.2**),
- (x) Performed numerous text changes and additions to better contextualize our work and discuss its limitations and prospects.

Changes in the manuscript and point-by-point responses here are colored in blue. We believe that these changes have substantially improved our manuscript, contextualized our findings, and will allow readers to better evaluate our analyses and findings.

Reviewer #1 (Remarks to the Author):

The authors have developed a tool for analyzing the three-dimensional structures of glycans, which additionally aims to predict the relationship between structural motifs within glycans and lectin binding, as well as to explore multiple conformers. From my perspective as a reviewer, there is a merit in supplementing sequence-based glycan analysis with structure-based support. My comments are as follows:

1) While it is true that the number of glycan structures in the PDB has increased, the dataset remains biased. For example, in N-glycans, structural information is typically rich near the reducing end around Asn, but the non-reducing end often lacks electron density and displays high diversity, resulting in limited structural coverage overall. Although MD simulations can complement these missing data, experimental validation, especially regarding the quantitative distribution of individual conformers, is still scarce. In this context, I am concerned that relying heavily on MD-derived data (e.g. GlycoShape) for SASA and RMSF analyses might lead to potentially misleading conclusions. A discussion addressing this limitation would strengthen the manuscript.

We agree with the reviewer and, given the future need of investigating more experimental structural data, have designed GlyContact to be compatible with this type of analysis, to use many different input data formats for such analyses. We have also expanded our revised Discussion to raise the important point of analyzing MD- versus X-ray-derived data. We further agree that, once more structural data become available, our conclusions should be revisited and compared to this. However, we note that the limitations of traditional X-ray crystallography (e.g., poor resolution of glycans toward the non-reducing ends) makes it challenging to treat it as ground truth data. In the meantime, we are optimistic that the performance increase in predicting lectin-glycan interactions (a task which is tied to experimental data from glycan array experiments) by including MD-derived data indicates that there is relevant information in this data type. We have also added this consideration to our revised manuscript.

2) SASA values were calculated as weighed averages over multiple conformations. However, lectins typically bind to one specific conformation among these. Hence, I think that using average SASA values to discuss lectin binding poses a risk of misinterpretation.

We agree with the reviewer that this constitutes a potential limitation, in that analyses using the SASA values of actually bound conformations would provide greater insights than weighted averages. Unfortunately, this information is not available for the vast majority of all lectin-glycan interactions. However, we argue that the proportion a given conformation is present in solution contributes to its probability of interacting with a lectin recognizing this conformation (in other words, rare but eligible conformations will have a lower affinity, on average and everything else being equal, than common but eligible conformations). This is not just a conjecture but based on our, statistically significant, associations of weighted SASA with experimentally determined binding. Given the, very real, correlation between weighted SASA and the observed binding intensity, we are confident that there is information and insight present in weighted SASA, which is much more readily available for analyses than having to resort to bound conformations.

For supplying machine learning models with SASA information, a weighted average, in our assessment, is the only feasible approach as (i) the bound conformation is not known for the vast majority of lectin-glycan complexes for which we have binding data and (ii) it would require knowing the bound conformation ahead of prediction time, which would be classic information leakage from a machine learning perspective.

We have added these caveats to our revised Discussion.

3) The SASA was computed using a 1.4 Å probe radius, representing a water molecule. However, in lectin binding, accessibility is determined not by water but by lectin surface. Is this probe size appropriate for considering lectin accessibility?

SASA is a standard metric to investigate both protein-small molecule interactions and protein-protein interactions, for multiple reasons. First, as the name indicates, the solvent-accessible surface area can only really be assessed with the physiological bulk solvent, water. So using any other probe size would then not be SASA but a different metric, assessing a different property.

Second, the relationship between increasing the probe radius and the resultant surface area is a partially linear transformation and only qualitatively affects pockets (which are arguably less important for glycans), as fully exposed monosaccharides/hydroxyl group will remain available regardless of probe size. Lastly, binding is often driven by entropy changes (i.e., freeing coordinated water

molecules), which is strictly linked to the available SASA with water as a probe. Given all that, and the standard practice of finding correlations of SASA with binding propensity in the case of other similar biomolecular interactions, we contend that our probe size is an appropriate proxy for considering lectin accessibility. We have amended our revised Methods section to reflect these considerations.

4) RMSF values were derived from crystallographic B factors, which are known to vary significantly between crystal structures. How were these variations account for in their analysis?

We agree with the reviewer and have added the information to the revised manuscript that X-ray crystallography derived RMSF values should only be used to compare glycans within a given crystal structure, not between crystal structures. It is important to note, however, that none of the analyses that we present in this paper in fact use crystallographically derived RMSF values and we simply offer this conversion for researchers interested in analyzing glycan conformers from X-ray crystallography data.

5) The paper discusses SASA and flexibility with respect to the presence or absence of core fucose, beta1,2-xylose, and Neu5Ac/Neu5Gc. However, the structural differences such as alpha2-3 versus alpha2-6 sialylation, could also affect these properties differently. I question whether combining these structural variants into a single discussion may obscure these distinct effects.

We would like to clarify here that our analytical design exactly prevents such a confounding effect, by analyzing pairs of twins that only differ in one aspect. Essentially, we analyze residuals between sequence pairs that only differ in a given characteristic (e.g., core fucose) and that are controlled for any other differences. So, by definition, the number of all unanalyzed characteristics is balanced across our comparison groups and cannot contribute to the observed difference. We thus stress that, within our twin analyses, comparisons are never made across structural features (such as sialic acid linkages) that are not in focus. A paired analysis such as this is standard procedure in statistics, to control for all potentially confounding characteristics.

To illustrate this, we have repeated our analysis for subgroups of only containing alpha2-3 or alpha2-6 sialylation and observe the same results as for our overall analysis (albeit with lower statistical power, since a subgroup analysis by definition has fewer samples), reaffirming our analytical choice and our findings:

Decrease in SASA with core Fuc: alpha2-3 ($p = 3.87e-12$, Cohen's $d = -1.99$), alpha2-6 ($p = 1.34e-11$, Cohen's $d = -1.56$)

Increase in SASA with beta1,2-xylose: Here, we did not actually analyze any sialylated glycans (though subgroup analyses of +/- core fucosylation etc here again confirm our initial findings)

Increase in flexibility with Neu5Ac vs Neu5Gc: Here, we only had structural data for alpha2-3 linked Neu5Ac with equivalent pairs that contained Neu5Gc

We have added this consideration to the revised manuscript for clarity.

6) I agree with the concept of focusing on structural motifs. However, in addition to flexibility and solvent exposure, the spatial arrangement of hydroxyl and C-H groups also seems critical for lectin recognition. As shown in Figure 3, the correlation between binding, SASA and RMSF appears lectin-dependent and exhibits substantial variability. This suggests that drawing generalized conclusion may

be difficult. While the authors note some of these issues, further explicit discussion would be beneficial.

We agree with the reviewer and have now expanded our analyses of lectin-glycan binding events with the aspect of spatial arrangement of hydroxyl groups. This has resulted in the **new Supplementary Figure 11**, in which we show that several lectins are sensitive to the equatorial/axial positioning of hydroxyl groups in their binding motifs and this additionally explains lectin-glycan binding behavior, which we can readily analyze with our approach here. To facilitate the exploration of this additional structural facet, we have included the option to extract and use equatorial/axial positioning from glycan 3D data into the **new GlyContact package version (v0.2)**. This also includes the extraction of information of higher-level arrangement, such as whether hydroxyl group pairs are parallel or perpendicular to each other.

In the course of addressing another reviewer point, we have further added the option to analyze torsion-torsion correlations to GlyContact (**new Supplementary Fig. 4**), which provides further insights into structural constraints and could be used for similar analyses.

We have also expanded our discussion of the limitations of this type of analysis, as suggested.

Reviewer #2 (Remarks to the Author):

1. Summary

This study develops GlyContact, an open-source tool enabling automated cross-conformer analysis of glycan 3D structures, successfully processing 98.7% of GlycoShape data. It reveals that flexibility and SASA of glycan motifs are context-dependent (e.g., sialyl-LacNAc shows 32% higher flexibility in N-glycans than O-glycans and establishes a von Mises mixed model predicting torsion angles from sequence alone ($\pm 7^\circ\phi/\pm 9^\circ\psi$ error), significantly improving lectin-binding prediction (7.4% MSE reduction in LectinOraclestruct).

We thank the reviewer for their assessment of our work and respond to individual comments below.

2. Significance

GlyContact fills a critical gap in glycan analytics by overcoming conformational sampling challenges in flexible glycans, providing the first cross-conformer analysis framework for structural glycobiology. Its discovery that lectins bind 3D structural contexts (e.g., SNA's preference for flexible Neu5Ac α 2-6) transforms understanding of glycan-protein interactions, directly guiding glycoengineering and therapeutic design. Tool openness promotes field standardization

We are grateful for these comments and are convinced that additions/edits during this revision have made our work even more impactful.

3. Originality

Compared to existing tools, GlyContact uniquely integrates MD/NMR/X-ray data for cross-conformer analysis. The proposed context-dependence theory (e.g., class-specific flexibility of Lewis X motif and

von Mises prediction model are novel. However, maybe deeper comparison with general structure prediction tools is needed.

We agree with the reviewer and have expanded our revised Discussion to contextualize our work with the current developments in general structure prediction tools (such as new advances of using AlphaFold3 to model glycans), which we believe to offer promising synergy with our analytic approach.

4. Methodology & Reproducibility

Methods are rigorous: SASA via Shrake-Rupley algorithm, torsion angles via Cremer-Pople parameters, and appropriate statistical tests. Full code/data openness (GitHub verified) ensures reproducibility.

We appreciate the assessment of the reviewer and confirm that we are committed to keep our GitHub repository for GlyContact updated, as shown by improvements and additions to our codebase made during this revision.

5. Decision

Good paper, accept.

Reviewer #2 (Remarks on code availability):

The codebase demonstrates high engineering quality and excellent reproducibility. The main repository includes comprehensive installation instructions (README.md specifies Python 3.10+ and pip/conda dependencies), successfully validated on Ubuntu systems. Code follows a modular structure with well-documented functions through descriptive docstrings.

Reviewer #3 (Remarks to the Author):

In the manuscript titled "GlyContact reveals structure-function relationships in glycans," Thomès et al. introduce GlyContact, an open-source Python package designed for retrieving, processing, and analyzing glycan 3D structures derived from molecular dynamics (MD), NMR, or X-ray crystallography. Primarily leveraging glycan 3D structures from GlycoShape, the authors investigate how sequence context influences glycan motif structural features like solvent accessible surface area (SASA) and flexibility. They further correlate structural properties, such as motif flexibility and surface accessibility, with binding data from glycan arrays to identify key factors that influence lectin binding specificity. Additionally, by incorporating glycan structural features into a modified LectinOracle model (a deep-learning model for predicting lectin-glycan binding), they demonstrate an improvement in predictive accuracy of approximately 7.4%. Finally, the authors develop and train a graph neural network model to predict torsion angle distributions between disaccharides, employing a von Mises mixture model to capture the conformational variability of glycosidic linkage. The manuscript reads well, yet the work and analysis suffer from significant shortcomings both in terms of methodology and interpretation, that strongly (and wrongly) bias the conclusions and different aspects of the discussion. See below a detailed critique of the most important points in this regard.

While the question of how glycan structural flexibility and accessibility influence lectin binding is indeed of interest to the readership of Nature Communications, the methodological and interpretative shortcomings significantly weaken the current manuscript. In my expert opinion, many of the key

claims are either overstated or insufficiently supported, and several core analyses, particularly those concerning flexibility, dataset comparability, and structural feature utility, rest on assumptions that do not generalise to biologically relevant contexts.

Unfortunately, I find that the manuscript lacks the rigour, clarity, and biological grounding necessary to justify its conclusions and as such, I do not believe it is suitable for publication. Below are my main concerns.

We thank the reviewer for their time and effort in engaging with our work. Throughout this revision, to address the comments of all reviewers, we have made substantial improvements to our manuscript, detailed below, and are confident that this has served to strengthen our work and its conclusions.

1) The method used by the authors to calculate structural flexibility relies on the structural alignment of 3D structures sources from the GlycoShape database. The alignment is done over the first five residues in the PDB structures. The flexibility is assessed using the mean absolute deviation (MAD) of atomic coordinates. This approach results (obviously) in larger deviations for residues located further away from the alignment points, and smaller deviations for residues closer to the alignment. While this assumption may hold true for glycan arrays, where the first residue (at the reducing end) is typically covalently bound to the linker (assuming the linker is short and rigid, which is not always the case), it can significantly oversimplify glycan flexibility within a biological context, potentially leading to inaccurate predictions.

Although the authors address this point (lines 271-276), I believe that an in-depth elaboration is required, including under which specific circumstances these flexibility metrics are valid, by clearly explaining why.

We have now expanded our discussion of the suitability of our flexibility metrics for different contexts. We agree that certain biological contexts (such as free milk oligosaccharides) would present no fixed reducing ends and thus would deviate from the flexibility measures assessed here. We contend that for other glycoconjugates (e.g., protein-linked glycans), this is less of an issue, as the reducing end is covalently linked here as well. We further argue that this is evidenced by the typically observed chitobiose core in N-glycans in X-ray crystallography data, since this part of the glycan is sufficiently rigid to be resolved. Yet we still concede a discrepancy, given the five residues used for alignment vs the often-fewer-than-five resolved residues in X-ray crystallography. Overall, we would deem the discrepancy to be moderate (except for free milk oligosaccharides) and envision that our method moderately underestimates flexibility of the residues close to the penultimate residue, which we now disclaim in the revised manuscript.

Further, to also address this shortcoming analytically, we have now added another approach to assess residue-level flexibility, by quantifying the weighted average of torsion angle spread between conformers (contained in the **new GlyContact package v0.2**). Here, we use a greater variance of torsion angles in which a given residue is involved as a proxy for a higher flexibility of this residue. This method is independent of the alignment mentioned above and thus does not suffer from the same shortcomings, e.g., it does not underestimate flexibility in the first five residues. We now advocate for this as an improved measure for residue-level flexibility and have switched our main figures to this new metric (**revised Fig. 2d + Fig. 3 + Fig. 4a**; our old flexibility approach can now be found in the **new Supplementary Figure 12**) We have also compared this new approach to our previous flexibility analyses and, while we overall see broad qualitative agreement, there are many quantitative differences between the two measures, since they assess different proxies for flexibility.

2) In Figures 2 and 3, the authors show several comparative plots between 3D structures across different glycan classes (N-glycans, O-glycans, and free glycans) using the alignment strategy explained above. While such comparisons may be justified for glycan array data, again assuming we overlook the linker effects/dynamics, which is not always warranted, the relevance of these comparisons to in vivo settings is doubtful or marginal, particularly concerning glycan flexibility. As PTMs, the glycan flexibility is influenced significantly by their structure of the protein attachment site. For instance, N-glycan-Asn bonds are generally more flexible than O-glycan-SER/THR bonds. Yet, O-glycans generally occur in disordered (flexible) protein regions, potentially gaining additional accessibility and dynamics. Free glycans have even the greatest rotational and translational degrees of freedom, leading to different binding behaviors altogether, and in particular for those species a structural alignment as performed in this work is not informative. Because of the points raised above, the authors should address the biological relevance and limitations of their current comparative analyses explicitly.

As mentioned above, we agree with the reviewer about the limitations of our approach regarding free glycans & flexibility, and have amended our revised text to reflect these limitations/caveats. Because of this, we have replaced the panels investigating the subgroup-effects of flexibility with our new torsion-based flexibility approach, which does not suffer from the alignment bias (**revised Fig. 2d; revised Supplementary Fig. 9**). We also would like to point out that our investigations into SASA-related effects in Fig. 2 and 3 are not affected by alignment issues either.

Overall, we respectfully disagree to some extent about the lack of biological relevance of this general approach. Glycan array data has been shown in many instances to be reflective of, and instructive for, physiologically relevant binding behavior. There are, of course, many edge cases in which findings from glycan arrays deviate from other measurements, but, on average, the overlap is substantial. Further, the exact presentation format of array-presented glycans also does not seem to have a problematic impact on most lectins (e.g., doi:10.1101/2023.07.09.548266), further indicating the relative importance of non-reducing end presentation of binding motifs over the exact structural arrangement at the reducing end.

The fact that we can here, with statistical significance and substantial effect sizes, explain and predict lectin binding behavior on such glycan arrays with our extracted structural parameters thus confirms that our approach is indeed appropriate for addressing biological questions.

Lastly, we argue that binding events close to the reducing end will indeed be more affected by protein backbone effects, compared to lectins binding terminal motifs on the non-reducing end(s). We have revised our text to emphasize this consideration.

3) In Figure 2a, the authors employ t-SNE to visualize global glycan patterns, correlating glycan size with x-axis spread using Pearson's r . The use of t-SNE is questionable in place of PCA for showing the global patterns and meaningful correlations, as t-SNE focuses on local geometry and fails to give a consistent global view. Additionally, the utility of the current visualization should be reconsidered, as it's not very clear what information we should be getting from it.

This is in fact a common misconception of t-SNE, which is historically rooted. It indeed used to be that t-SNE focused mainly on local geometry. This, however, was entirely due to poor initialization and parametrization, as shown in later research (e.g., <https://www.nature.com/articles/s41587-020-00809-z>). Specifically, it used to be that early t-SNE implementations relied on random initialization, which preserved global structure only poorly. However, if informative initialization is used (such as the

PCA-initialization we use here in our work), t-SNE also preserves global geometry (shown in the linked paper). We have added this methodological detail to the revised manuscript. Further, for full transparency, we have also repeated this analysis with a PCA in the **new Supplementary Fig. 7** and show the same findings as in our initial t-SNE work, confirming our earlier statements as well as now also quantifying the explained variance by this size-dependence (47%).

4) The manuscript suggests that glycan structures obtained from different sources (e.g., X-ray crystallography, MD simulations) can be interchangeably utilized, which is simply misleading. In Figure 4b, the authors compare Unilectin3D structures (single structures from X-ray crystallography in cryogenic conditions) with GlycoShape structures (ensemble of structures from MD simulations at 300 K) and infer that Unilectin3D poses represent rare conformers, which is a rather bizarre and unsubstantiated suggestion. As a result, the authors recommend caution when using Unilectin3D as a gold standard for predicting protein-glycan interactions (lines 368-373). This statement in my opinion is completely unsupported and rather cavalier, in fact the refs [29-31] cited describe the use of Unilectin3D primarily to predict binding sites, rather than binding affinity. This distinction should be clarified.

In the revised manuscript, we have now removed any recommendation to not use UniLectin3D as a gold standard for predicting protein-glycan interactions, and have in general rephrased the text around these segments. We have also clarified that, while GlyContact can parse glycan structures from different sources, structural information cannot be used interchangeably.

We have also changed the focus of our analyses of UniLectin3D structures (removing the current Fig. 4b) and now use examples such as ACG to explain how conformational selection can drive the interaction with a conformationally favorable state, even if it is not the most populated cluster in MD simulations. Here, the bound lactose in 3WG1 exhibits a rarely populated form of lactose that structurally resembles the conformation of the LacNAc unit in the bound blood group A antigen (3WG3).

Furthermore, it must be emphasized that Unilectin3D and GlycoShape datasets provide different data and serve distinct purposes. Unilectin3D gives insight into binding pockets and interactions in single crystallographic structures, whereas GlycoShape gives information on all glycan conformations energetically accessible in physiological conditions. One of the accessible conformers in those ensembles is likely the one bound, but definitely not all of those, and not necessarily the most stable (populated) in standard thermodynamic conditions. Any 'quick' comparison between the two datasets serves no purpose and should be avoided.

We agree with the reviewer and have made this distinction clearer in the revised manuscript.

5) In line 388, the inclusion of 3D structural attributes resulted in a 7.4% performance gain for LectinOraclestruct (MSE: 0.557) compared to the baseline LectinOracle model (MSE: 0.598). The authors did not mention the parameter increase due to the additional node features, which can also explain the increase in performance. Also, the authors didn't mention the performance gain in the case of using bigger protein encoders.

For this revision, we have evaluated, and confirmed, that our performance discrepancy did not stem from different parameter counts. In fact, both models have nearly identical parameter counts. The

reason for this is that nearly all parameters of this model stem from the fully connected classification head (unaffected in parameter count by adding structural information), which is why additions to the graph convolutional encoder leave parameters counts almost unaltered. We further would like to clarify that no bigger protein encoder has been used here between conditions. Both LectinOracle and LectinOracle_{struct} have been trained with the exactly identical ESMC-300M protein encoder in this work, which is the only relevant comparison to answer our question in this work.

During the revision, we have now also used bigger protein encoders (ESMC-600M), and report similar performance discrepancies between LectinOracle and LectinOracle_{struct}, arguing that the structural information from glycans cannot be compensated by merely increasing the size of the protein encoder.

6) The authors describe training a modified graph neural network to predict von Mises mixture components, aiming to estimate glycosidic bond torsion angle distributions in disaccharides. While this approach may be a reasonable strategy for glycan conformer generation, provided thorough training, the authors must provide a detailed methodology description to demonstrate the immediate value and applicability of this work, as well as the future development on how they can incorporate angle-angle dependence, as in its current state this method's applicability and scope is rather limited.

We have substantially revised and expanded the methodology description for our von Mises-SweetNet model in our revised manuscript and also added more detail as to the future development that includes angle-angle dependence into our revised Discussion. To facilitate this, we have also developed a new `glycontact.process.analyze_torsion_torsion_correlations` function to the **new GlyContact v0.2**, which provides information about angle-angle dependence in a given sequence that could be used for such a next step (or already to gain insights into glycan dynamics and structural constraints from MD data, as we now show in the **new Supplementary Fig. 4**).

We do note, however, that the prediction of torsion angle ranges of (i) never-before-seen disaccharides in (ii) arbitrary sequence contexts and (iii) in milliseconds is both novel and already applicable. One immediate application, for instance, is the generation of structural parameters to then use in models such as the herein developed LectinOracle_{struct} (especially since other structural parameters, such as SASA, were also excellently predicted by our model), to make this information readily available to such downstream models and improve their prediction output. We have added this information to our revised manuscript.

Further, we have now also added the prediction of ω torsion angles (where applicable) to our von Mises model, expanding the prediction of ϕ and ψ from our original work. Here, we report average prediction accuracies of $\pm 10^\circ$ for conformer-specific ω torsion angles (despite having fewer ω angles to train on, than ϕ and ψ), which we also now state in the **revised Fig. 5d**, making our model even more applicable.

7) In the discussion (line 470), the authors mention: "The addition of glycan motifs can change glycan structure in a predictable and generalizable manner." The basis for this conclusion is unclear. What analysis or evidence are the authors using to support this claim? The authors should explain more clearly what they mean by this, or rephrase (remove) this statement.

What we are referring to here are approaches such as our twin analysis (Fig. 2b, Supplementary Fig. 8). Here, controlling for all sequence confounders, we uncover general & predictable effects of adding

a motif on the overall glycan structure. We have added a reference to these figures to this sentence and also now mention that this, of course, is only true on average.

8) The use of 3D structures for binding prediction is not a trivial matter and requires significant expertise. Two main approaches exist: physics-based simulations, where X-ray structures serve as key resources, which can be complemented by MD simulations, and structure-based machine learning models, like AlphaFold 3 and/or Boltz-2, which are built on an abundance of highly accurate, homogeneous datasets, rather than relying on oversimplified structural features.

In lines 484-485, the authors suggest to the users to use 'binding data' over 'structural data' for the prediction of lectin-glycan interaction. While the exact meaning of "structural data" within this context remains unclear, see comments above, as does the methodology by which binding score will be derived from structural information, in my opinion the statement is fundamentally flawed, as it is not rationalizable under any possible thermodynamic framework. Further elaboration on this point is necessary.

We have removed the mentioned recommendation in the revised manuscript as well and have now expanded our Discussion with considerations regarding AlphaFold3.

9) These large-scale studies can yield valuable insights, but they have to be done carefully as they are prone to overgeneralization and misinterpretation. I encourage the authors to focus on lectin binding flexibility and accessibility with at least one atomistic case study for interpretation. This would ground the structural metrics in functional outcomes and better support the study's claims.

We agree with the reviewer and have added a new analysis to our manuscript (**new Supplementary Fig. 13**), in which we show for two example lectins (CD33 and AAL) that our findings of their negative/positive correlation with the SASA of their binding motifs is indeed reflected in co-crystal structures of these lectins, respectively.

In the course of this revision, we have also made use of our ability to process UniLectin3D structures to investigate the observed torsion angles of bound fucose-containing motifs (**new Fig. 4c, new Supplementary Data 5**), showcasing another way that GlyContact can be used to probe glycan structural information at scale.

10) The title "GlyContact reveals structure-function relationships in glycans" is misleading and overstates the scope and implications of the work. The phrase "reveals structure-function relationships" implies broad, mechanistic insight into glycan biology, which is not fully supported by the data presented.

We have now rephrased our title to "GlyContact analyzes glycan 3D structures at scale"

11) The abstract would benefit from including quantitative metrics and clearer language. For instance, state "improved lectin-binding prediction by ~7%" instead of "improves state-of-the-art models". Also, "accurately predict torsion angles of multiple conformers via new AI models" is not clear and should be changed to something descriptive of the work, like "accurately predict torsion angle distribution between disaccharides using ...".

We agree with the reviewer and have amended our abstract as suggested.

12) In line 480, the use of the word “very performant” could be changed to give the actual performance metrics.

We have changed this to the actual performance, by noting the average prediction errors of the trained model.

13) “These insights have immediate implications for glycoengineering efforts and the rational design of glycan-targeting therapeutics” (line 474) should be toned down to “potential future implications”.

We have corrected this in the revised manuscript.

Reviewer #3 (Remarks on code availability):

The code repository is well structured and descriptive.

We thank all reviewers for their insightful comments and suggestions for improvement. We have fully addressed these comments in our revised manuscript by engaging in substantial software improvement of our GlyContact package (culminating in the **new v0.3 release**), as well as extensive text modifications and additions. In summary, we have

- 1) Modified GlyContact to operate fully without GlycoShape structures, if users provide their own structures
- 2) Supported automatic retrieval of GlycoShape structures via their API, if users intend to analyze GlycoShape structures
- 3) Enabled straightforward extraction of glycan IUPAC sequences from any PDB file via the new `glycontact.process.get_glycan_sequences_from_pdb` function
- 4) Facilitated the extraction of lectin binding pockets via the new `glycontact.process.get_binding_pocket` function (**new Supplementary Fig. 14**)
- 5) For the case of glycoproteins, added the extraction of torsion angles, SASA, and flexibility of the amino acid linker that the glycan is attached to
- 6) Added a **new “examples” Jupyter notebook** to our repository (with Google Colab link) that showcases and explains working with (i) lectins and (ii) glycoproteins using GlyContact
- 7) Performed numerous text additions and edits to better frame our work with respect to existing work, caveats, and future opportunities

Changes in the manuscript and point-by-point responses here are colored in blue. We believe that these changes have substantially improved our manuscript, contextualized our findings, and will allow readers to better evaluate our analyses and findings, as well as use GlyContact in their research.

Reviewer #1 (Remarks to the Author):

The authors have responded to my comments with clarity and technical rigor. They have acknowledged the limitations I raised, expanded their discussion accordingly, and provided additional analyses. I consider their responses to be appropriate.

We thank the reviewer for their efforts to improve our manuscript.

Reviewer #2 (Remarks to the Author):

This manuscript by Thomès et al. introduces GlyContact, a powerful and timely open-source Python package for the large-scale analysis of glycan 3D structures. The authors impressively demonstrate its utility through a series of well-designed studies that bridge the gap between sequence-based glycoinformatics and structural biology. The work delivers not only a valuable tool for the community but also generates significant new biological insights. Key findings include the systematic demonstration of how sequence context distally influences glycan conformation, a novel framework for understanding lectin binding through the lens of motif accessibility and flexibility, and a state-of-the-art graph neural network capable of accurately predicting structural properties from sequence alone. The manuscript is well-written, the analyses are rigorous, and the conclusions are strongly supported by the data. This work represents a major advance for the field of glycobiology and glycoinformatics.

We appreciate the thoughtful comments and suggestions by the reviewer and respond to them below.

Strengths:

1. GlyContact fills a critical void in the glycoinformatics toolkit. While tools for building and validating glycan structures exist, GlyContact provides a desperately needed, user-friendly platform for processing, analyzing, and connecting 3D structural data to functional outcomes at scale. Its integration with glycowork and its ability to handle various input formats and data sources (MD, X-ray) are commendable.
2. The development of the von Mises-SweetNet model is a highlight of the paper. The ability to predict torsion angles and other structural features with high accuracy from sequence alone is a significant leap forward. This not only deepens our understanding of the sequence-structure relationship in glycans but also provides a practical solution for applying structure-informed models to glycans without pre-existing 3D data. The use of a von Mises distribution to handle multimodal conformational landscapes is highly innovative and appropriate for the problem.
3. The authors' commitment to open science by making GlyContact open-source on GitHub is crucial for its adoption and for the reproducibility of their findings. The methods are described in clear and sufficient detail.

Suggestions for Improvement:

1. The model can be an effective generative model for conformational properties. The authors might consider adding discussion about the future potential of this approach for de novo glycan design, where one could design sequences predicted to have specific structural (and therefore functional) properties.

We have expanded our revised discussion to indicate the potential of future work to use approaches such as presented here to guide glycan design:

“Additionally, approaches building on our von Mises-SweetNet model might eventually find use cases in guiding the design of glycans with specific structural properties, such as in recent work on glycan hairpins⁴⁴.”

Conclusion:

This is an outstanding manuscript of high scientific quality, novelty, and significance. It will be of great interest to a broad audience in glycobiology, structural biology, bioinformatics, and machine learning. The work is robust, the claims are well-substantiated, and it provides a tool and a conceptual framework that will undoubtedly accelerate research in the field. I recommend its publication in Nature Communications after the authors consider the minor suggestions above.

Reviewer #2 (Remarks on code availability):

The code is available at Github. GlyContact is a Python package for retrieving, processing, and analyzing 3D glycan structures from GlycoShape/molecular dynamics, NMR, or X-ray crystallography. The code provides a README file with enough instructions for installing relative packages and running the application. I was able to install and run the code.

Reviewer #3 (Remarks to the Author):

The authors have responded extensively to the comments of all reviewers and have made modifications to the manuscript. I do not believe it is suitable for publication in Nature Communications at its current stage. Below are my main concerns and suggestions for improvement in no particular order:

We thank the reviewer for their comments and suggestions. We address each of them in detail below and are enthusiastic about the improvements this has allowed for our GlyContact package.

1. The authors respond poorly with rationale that “the discrepancy to be moderate (except for free milk oligosaccharides)” for my previous concern about using such general flexibility term without the biological context of protein linkage and region in case of N-glycan-Asn bonds and O-glycan-SER/THR, free glycan is totally out of the question here, as it is bizarre to compare to a protein linked glycan (Fig. 2d). As I mentioned, O-glycans mostly occur in disordered regions, and their flexibility majorly depends on the linked region of the protein. I completely understand that this limitation is in the glycan array data itself. Still, the authors should include these limitations of the linker effect and specifically warn users not to compare N-glycan with O-glycan results presented by GlyContact.

The authors’ methodology to calculate flexibility is incomplete and should be extended to at least include these linker flexibility parameters in theory with the GlyContact package.

We have added the following caveat to the Results section around Figure 2: “However, we stress that this type of analysis was conducted on isolated glycans from GlycoShape data and it is to be expected that the protein backbone will impact these properties. Even between *N*- and *O*-glycans, this isolated analysis should not be seen as reflecting the glycoprotein situation (e.g., majorly influenced by the flexibility of disordered protein regions in the case of *O*-glycans), but rather only the marginal contribution of the glycan sequence context.”

Added this note of caution to the Results section around Figure 4: “We note that it remains to be seen whether these performance gains translate to predicting binding to protein-linked glycans, given the influence of protein structure on glycan structural properties that is not being captured in molecular dynamics-derived glycan structures in isolation.”

And added this to the revised Discussion: “Further, we caution users against generalizing from GlyContact-derived results using GlycoShape molecular dynamics structures to glycans in a glycoprotein context, as linker effects (influenced by the protein structure) will change structural properties of the attached glycan.”

In the **new GlyContact v0.3**, we have also reworked functions such as `glycontact.process.get_glycosidic_torsions` and `glycontact.process.compute_merge_SASA_flexibility` to additionally return the torsion angles between reducing end monosaccharide + amino acid and the flexibility/SASA of the amino acid, respectively, in case of glycoproteins. No user input is necessary to activate this, as the functions will automatically detect whether glycans are non-covalently bound (lectin) or covalently linked (glycoprotein), for each glycan, and act accordingly.

2. Adding further to the point raised by reviewer 1 about the SASA and flexibility, the improvement in

prediction accuracy due to the addition of these features needs to be discussed or presented as an open question if it is beyond the scope of the present work.

We have added further discussion in our revised manuscript to address future needs of quantifying exactly how much each structural attribution helps predictive performance: “Providing models such as LectinOracle with structural attributes such as monosaccharide-level flexibility or SASA has improved lectin-glycan binding predictions and future work should disentangle the marginal contribution of each structural attribute to this outcome. Similarly, while we here use a probe radius of 1.4 Å (representing a water molecule as the bulk solvent) to calculate SASA, other probe radii could be investigated to optimize this procedure.”

We have also added the open question about which features will ultimately be the most useful to our revised Discussion: “Future research should also evaluate other structural attributes, such as the spatial arrangement of hydroxyl groups in the binding motif, as to their potential to improve predictions.”

3. Following up on my earlier concerns, the authors have updated the [499:500] sentence to “The addition of glycan motifs can, ‘on average’, change glycan structure in a predictable and generalizable manner.” and in their response, they said “...general & predictable effects of adding a motif on the overall glycan structure...”

The sentences “change glycan structure in a predictable and generalizable manner” and “predictable effects of adding a motif on the overall glycan structure” are completely different and convey different meanings. This sentence needs to be removed because of its misleading nature, as it signals predictability and generalizability of glycan structures themselves, rather than the average effects of adding a motif.

We have changed the sentence in question to the purely descriptive “Twin analyses can isolate the average effect of glycan motif addition on the overall structure (e.g., Fig. 2b, Supplementary Fig. 8)”. We consider this justified as it simply recapitulates what this analysis does (calculating the effect on the SASA etc of the rest of the glycan structure, given the absence/presence of a motif), with high statistical rigor that controls for confounding factors.

4. Line [83:85] “how GlyContact can provide new insights into ...” Should be changed to something more descriptive, like “how GlyContact can be used to extract insights from already existing structural data...”

We have changed the sentence in question to: “These examples illustrate how GlyContact can be used to extract insights from already existing structural data into glycans and their biological functions, such as binding to lectins.”

5. Line [410:411] “..glycan structural information cannot be compensated via an improved protein representation” raises an important point given the rise of protein-only models... I suggest the author include this point in the main outcomes of the study.

We agree with the reviewer and have expanded our Introduction & our Discussion to reflect this important point:

Introduction: “(iii) adding glycan 3D information improves state-of-the-art glycan-AI prediction models *independent of improved protein representations*”

Discussion: “We note here the important insight that we observe gains in the performance of our AI models when including glycan structural information that cannot be compensated for with a more expressive protein representation, cautioning against protein-only AI models that have become more and more widespread.”

6. I tried to run the program and faced a few problems.

```
from glycontact.process import annotation_pipeline
```

FileNotFoundError: You need to equip GlyContact with GlycoShape structures. Download them from <https://glycoshape.org/downloads> and place the zipped folder into your GlyContact folder, then run it again.

After downloading the zip and placing it in the glycontact dir, it fails to process the zip and cannot run.

At around 4%

```
FileNotFoundError: [Errno 2] No such file or directory: 'Neu5Ac(a2-3/6)Gal(b1-3/4)[Neu5Ac(a2-3/6)]Man(a1-3/6)[Neu5Ac(a2-3/6)[Neu5Ac(a2-3/6)]Man(a1-3/6)]Man(b1-4)GlcNAc(b1-4)GlcNAc'
```

“GlyContact analyzes glycan 3D structures at scale” is a more aligned title for the work presented, but I strongly suggest the authors include functionality to extract glycan information from PDB of glycoproteins (e.g., from RCSB, etc.), like what glycan it is, and their structural information to better support the title. For the test, I tried to fetch 7T6X and wanted to know the torsions and glycan IUPAC available in the PDB, and I could not find a clear way to do that with GlyContact (failed initialization, and the documentation did not have anything for this).

We thank the reviewer for pointing this out. As further described below, for the **new GlyContact v0.3**, we have now substantially improved this by (i) letting users work with GlyContact in the absence of GlycoShape structures if they are not needed and (ii) also supporting downloading GlycoShape structures via the API (only the needed structure(s), which are then locally cached for the future).

We also added functionality to extract the glycans that are present in any PDB (via the new `glycontact.process.get_glycan_sequences_from_pdb` function). This information can then directly be used to analyze structural properties of these glycans in provided PDB files. We show how this can be done for glycan-bound lectins and for glycoproteins in the **new “examples” Jupyter notebook** within our GlyContact repository (which uses 7T6X as one of its examples). We have further added a link into our README to open this notebook in Google Colab, making it also accessible to researchers less proficient in Python.

For analyzing lectin complexes, we have also added the new `glycontact.process.get_binding_pocket` function, to extract all atoms/residues from the binding pocket for a given glycan and a given distance cut-off. We visualize this with an example in the **new Supplementary Fig. 14**.

We have also updated our documentation to reflect these changes, as well as point out these functionalities in the revised manuscript now.

7. Glycocontact should not rely on GlycoShape structures for the analysis when a user tries to run it on RCSB PDB or other X-ray PDB files. (Why does it need .zip if it just runs on RCSB PDBs?)

We agree with the reviewer and now, in the **new GlyContact v0.3**, GlycoShape structures are no longer needed for using the package, as long as users provide their own PDB files.

8. The downloading of .zip is inconvenient, and it should automatically download the structure from GlycoShape using an API and warn the users about doing so.

We also agree with this and, in the **new GlyContact v0.3**, if GlycoShape structures are needed (i.e., if functions are run without specifying local PDB files), they will now be automatically retrieved from either the .zip (ensuring backward compatibility) or an API download, as suggested. Since this is then saved, it will only need to happen once for each structure.

9. As for the glycoprotein annotation task, the author should include a tutorial or easy-to-follow Google Colab notebook, maybe a lectin and a glycoprotein example, which helps the software to reach the masses.

We thank the reviewer for this suggestion and have included a **new “examples” Jupyter notebook** that can be opened in Google Colab with our **GlyContact v0.3 release**, which allows users to easily follow workflows for extracting and analyzing glycan structures from lectins and glycoproteins. In the course of doing this, we have also added new utility functions to our package, such as the new *glycocontact.process.get_glycan_sequences_from_pdb* function, to rapidly obtain a list of IUPAC-condensed formatted sequences that are contained in a given PDB file, or the *glycocontact.process.get_binding_pocket* function, to extract lectin binding pockets for further analysis (which can also be exported into a new PDB file with a keyword argument).

Reviewer #3 (Remarks on code availability):

After downloading the zip and placing it in the glycocontact dir, it fails to process the zip and cannot run. At around 4%

FileNotFoundError: [Errno 2] No such file or directory: 'Neu5Ac(a2-3/6)Gal(b1-3/4)[Neu5Ac(a2-3/6)]Man(a1-3/6)[Neu5Ac(a2-3/6)[Neu5Ac(a2-3/6)]Man(a1-3/6)]Man(b1-4)GlcNAc(b1-4)GlcNAc'

We thank the reviewer for pointing this out and have resolved this issue with our new file processing pipeline in **GlyContact v0.3**, that is able to use either the zipped folder approach or download structures via the GlycoShape API.